# CROSS-MODAL CONTEXTUALIZED DIFFUSION MODELS FOR TEXT-GUIDED VISUAL GENERATION AND EDITING

**Ling Yang**[1*†]   **Zhilong Zhang**[1*]   **Zhaochen Yu**[1*]   **Jingwei Liu**[1]   **Minkai Xu**[2]
**Stefano Ermon**[2]   **Bin Cui**[1†]
[1]Peking University   [2]Stanford University
{yangling0818, bityzcedu, jingweiliu1996}@163.com
{minkai, ermon}@cs.stanford.edu,{zzl2018math, bin.cui}@pku.edu.cn

## ABSTRACT

Conditional diffusion models have exhibited superior performance in high-fidelity text-guided visual generation and editing. Nevertheless, prevailing text-guided visual diffusion models primarily focus on incorporating text-visual relationships exclusively into the reverse process, often disregarding their relevance in the forward process. This inconsistency between forward and reverse processes may limit the precise conveyance of textual semantics in visual synthesis results. To address this issue, we propose a novel and general contextualized diffusion model (CONTEXTDIFF) by incorporating the cross-modal context encompassing interactions and alignments between text condition and visual sample into forward and reverse processes. We propagate this context to all timesteps in the two processes to adapt their trajectories, thereby facilitating cross-modal conditional modeling. We generalize our contextualized diffusion to both DDPMs and DDIMs with theoretical derivations, and demonstrate the effectiveness of our model in evaluations with two challenging tasks: text-to-image generation, and text-to-video editing. In each task, our CONTEXTDIFF achieves new state-of-the-art performance, significantly enhancing the semantic alignment between text condition and generated samples, as evidenced by quantitative and qualitative evaluations. Our code is available at https://github.com/YangLing0818/ContextDiff

## 1 INTRODUCTION

Diffusion models (Yang et al., 2023b) have made remarkable progress in visual generation and editing. They are first introduced by Sohl-Dickstein et al. (2015) and then improved by Song & Ermon (2019) and Ho et al. (2020), and can now generate samples with unprecedented quality and diversity (Rombach et al., 2022; Yang et al., 2023a; Podell et al., 2023; Yang et al., 2024a). As a powerful representation space for multi-modal data, CLIP latent space (Radford et al., 2021) is widely used by diffusion models to semantically modify images/videos by moving in the direction of any encoded text condition for controllable text-guided visual synthesis (Yang et al., 2024b; Zhang et al., 2024; Ramesh et al., 2022; Saharia et al., 2022b; Wu et al., 2022; Khachatryan et al., 2023).

Generally, text-guided visual diffusion models gradually disrupt visual input by adding noise through a fixed forward process, and learn its reverse process to generate samples from noise in a denoising way by incorporating clip text embedding. For example, text-to-image diffusion models usually estimate the similarity between text and noisy data to guide pretrained unconditional DDPMs (Dhariwal & Nichol, 2021; Nichol et al., 2022a), or directly train a conditional DDPM from scratch by incorporating text into the function approximator of the reverse process (Rombach et al., 2022; Ramesh et al., 2022). Text-to-video diffusion models mainly build upon pretrained DDPMs, and extend them with designed temporal modules (*e.g.*, spatio-temporal attention) and DDIM Song et al. (2020a) inversion for both temporal and structural consistency (Wu et al., 2022; Qi et al., 2023).

---

[*]Contributed equally.
[†]Corresponding authors.

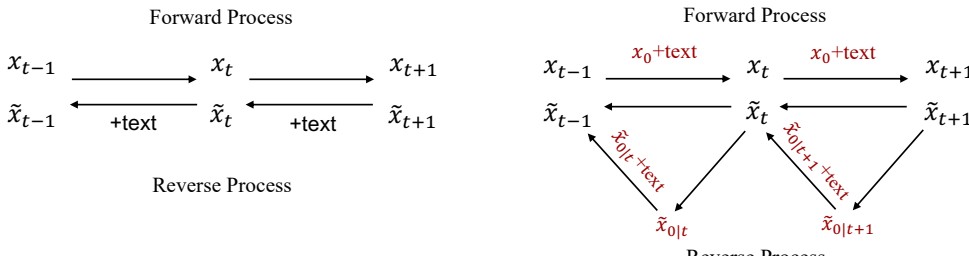

(a) Conventional forward and reverse diffusion processes    (b) Contextualized forward and reverse diffusion processes

Figure 1: A simplified illustration of text-guided visual diffusion models with (a) conventional forward and reverse diffusion processes, (b) our contextualized forward and reverse diffusion processes. $\tilde{x}_0$ denotes the estimation of visual sample by the denoising network at each timestep.

Despite all this progress, there are common limitations in the majority of existing text-guided visual diffusion models. They typically employ an unconditional forward process but rely on a text-conditional reverse process for denoising and sample generation. This inconsistency in the utilization of text condition between forward and reverse processes would constrain the potential of conditional diffusion models. Furthermore, they usually neglect the cross-modal context, which encompasses the interaction and alignment between textual and visual modalities in the diffusion process, which may limit the precise expression of textual semantics in visual synthesis results.

To address these limitations, we propose a novel and general cross-modal contextualized diffusion model (CONTEXTDIFF) that harnesses cross-modal context to facilitate the learning capacity of cross-modal diffusion models. As illustrated in Figure 1, we compare our contextualized diffusion models with conventional text-guided diffusion models. We incorporate the cross-modal interactions between text condition and image/video sample into the forward process, serving as a context-aware adapter to optimize diffusion trajectories. Furthermore, to facilitate the conditional modeling in the reverse process and align it with the adapted forward process, we also use the context-aware adapter to adapt the sampling trajectories. In contrast to traditional textual guidance employed for visual sampling process (Rombach et al., 2022; Saharia et al., 2022b), our CONTEXTDIFF offers a distinct approach by providing enhanced and contextually informed guidance for visual sampling. We generalize our contextualized diffusion to both DDPMs and DDIMs for benefiting both cross-modal generation and editing tasks, and provide detailed theoretical derivations. We demonstrate the effectiveness of our CONTEXTDIFF in two challenging text-guided visual synthesis tasks: text-to-video generation and text-to-video editing. Empirical results reveal that our contextualized diffusion models can consistently improve the semantic alignment between text conditions and synthesis results over existing diffusion models in both tasks.

To summarize, we have made the following contributions: **(i)** To the best of our knowledge, We for the first time propose CONTEXTDIFF to consider cross-modal interactions as context-aware trajectory adapter to contextualize both forward and sampling processes in text-guided visual diffusion models. **(ii)** We generalize our contextualized diffusion to DDPMs and DDIMs with thereotical derivations for benefiting both cross-modal visual generation and editing tasks. **(iii)** Our CONTEXTDIFF achieves new state-of-the-art performance on text-to-image generation and text-to-video editing tasks, consistently demonstrating the superiority of our CONTEXTDIFF over existing diffusion models with both quantitative and qualitative comparisons.

## 2   RELATED WORK

**Text-Guided Visual Diffusion Models**   Text-to-image diffusion models (Yang et al., 2023a; Podell et al., 2023) mainly incorporate the text semantics into the image sampling process (Nichol et al., 2022a) for cross-modal comprehension. Latent Diffusion Models (LDMs) (Rombach et al., 2022) apply diffusion models on the latent space of powerful pretrained autoencoders for high-resolution synthesis. RPG (Yang et al., 2024b) proposes a LLM-grounded prompt decomposition and utilizes the multimodal chain-of-thought reasoning ability of MLLMs to enable complex/compositional image generation. Regarding text-to-video diffusion models, recent methods mainly leverage the pretrained text-to-image diffusion models in zero-shot (Qi et al., 2023; Wang et al., 2023b) and one-shot (Wu et al., 2022; Liu et al., 2023) methodologies for text-to-video editing. For example, Tune-A-

Video (Wu et al., 2022) employs DDIM (Song et al., 2020a) inversion to provide structural guidance for sampling, and proposes efficient attention tuning for improving temporal consistency. FateZero (Qi et al., 2023) fuses the attention maps in the inversion process and generation process to preserve the motion and structure consistency during editing. In this work, we for the first time improve both text-to-image and text-to-video diffusion models with a general context-aware trajectory adapter.

**Diffusion Trajectory Optimization** Our work focuses on optimizing the diffusion trajectories that denotes the distribution of the entire diffusion process. Some methods modify the forward process with a carefully-designed transition kernel or a new data-dependent initialization distribution (Liu et al., 2022; Dockhorn et al., 2021; Lee et al., 2021; Karras et al., 2022). For example, Rectified Flow (Liu et al., 2022) learns a straight path connecting the data distribution and prior distribution. Grad-TTS (Popov et al., 2021) and PriorGrad (Lee et al., 2021) introduce conditional forward process with data-dependent priors for audio diffusion models. Other methods mainly parameterize the forward process with additional neural networks (Zhang & Chen, 2021; Kim et al., 2022; Kingma et al., 2021). VDM (Kingma et al., 2021) parameterizes the noise schedule with a monotonic neural network, which is jointly trained with the denoising network. However, these methods only utilize unimodal information in forward process (Yang et al., 2024a), and thus are inadequate for handling complex multimodal synthesis tasks. In contrast, our CONTEXTDIFF for the first time incorporates cross-modal context into the diffusion process for improving text-guided visual synthesis, which is more informative and contextual guidance compared to text guidance.

## 3 PRELIMINARIES

**Denoising Diffusion Probabilistic Models** Diffusion models (Ho et al., 2020; Song et al., 2020b) consider an unconditional forward process that gradually disturb the data distribution $q(\boldsymbol{x}_0)$ into a tractable prior $\mathcal{N}(\mathbf{0}, \boldsymbol{I})$ with a gaussian kernel defined by $\{\beta_1, \beta_2, \ldots, \beta_T\}$: $q(\boldsymbol{x}_t|\boldsymbol{x}_{t-1}) = \mathcal{N}(\sqrt{(1-\beta_t)}\boldsymbol{x}_{t-1}, \beta_t\boldsymbol{I})$, which admits a close form of conditional distribution of $\boldsymbol{x}_t$ given $\boldsymbol{x}_0$: $q(\boldsymbol{x}_t|\boldsymbol{x}_0) = \mathcal{N}(\sqrt{\bar{\alpha}_t}\boldsymbol{x}_0, (1-\bar{\alpha}_t)\boldsymbol{I})$, where $\bar{\alpha}_t = \prod_{i=1}^t (1-\beta_i)$. Then a parameterized Markov chain $\{p_\theta(\boldsymbol{x}_{t-1}|\boldsymbol{x}_t)\}_{t=1}^T$ is trained match the distribution of the reversal of the forward process. The training objective is a variational bound of the negative log likelihood of the data distribution $q(\boldsymbol{x}_0)$:

$$\mathcal{L} = \mathbb{E}_q[\log \frac{q(\boldsymbol{x}_{1:T}|\boldsymbol{x}_0, \boldsymbol{c})}{p_\theta(\boldsymbol{x}_{0:T}|c)}] \geq \mathbb{E}_q - \log p_\theta(\boldsymbol{x}_0). \tag{1}$$

$q(\boldsymbol{x}_{t-1}|\boldsymbol{x}_t, \boldsymbol{x}_0)$ admits a closed form gaussian distribution with the mean determined by $\boldsymbol{x}_0$ and $\boldsymbol{x}_t$, then $p_\theta(\boldsymbol{x}_{t-1}|\boldsymbol{x}_t)$ can be parameterized to gaussian kernel which mean is predict by $\boldsymbol{x}_t$.

**Denoising Diffusion Implicit Models** DDIMs generalize the forward process of DDPMs to non-Markovian process with an equivalent objective for training. Deterministic DDIM sampling (Song et al., 2020a) is one of ODE-based sampling methods (Lu et al., 2022; Song et al., 2020b) to generate samples starting from $x_T \sim \mathcal{N}(\mathbf{0}, \boldsymbol{I})$ via the following iteration rule:

$$x_{t-1} = \sqrt{\alpha_{t-1}}\frac{\boldsymbol{x}_t - \sqrt{1-\alpha_t}\epsilon_\theta(\boldsymbol{x}_t, t)}{\sqrt{\alpha_t}} + \sqrt{1-\alpha_{t-1}}\epsilon_\theta(\boldsymbol{x}_t, t). \tag{2}$$

DDIM inversion (Song et al., 2020a) can convert a real image $x_0$ to related inversion noise by reversing the above process, which can be reconstructed by DDIM sampling. It is usually adopted in editing task (Hertz et al., 2023; Mokady et al., 2022; Tumanyan et al., 2022; Qi et al., 2023).

## 4 METHOD

### 4.1 CROSS-MODAL CONTEXTUALIZED DIFFUSION

We aim to incorporate cross-modal context of each text-image(video) pair $(\boldsymbol{c}, \boldsymbol{x}_0)$ into the diffusion process as in Figure 2. We use clip encoders to extract the embeddings of each pair, and adopt an relational network (*e.g.*, cross attention) to model the interactions and alignments between the two modalities as cross-modal context. This context is then propagated to all timesteps of the diffusion process as a bias term (we highlight the critical parts of our CONTEXTDIFF in brown):

$$q_\phi(\boldsymbol{x}_t|\boldsymbol{x}_0, \boldsymbol{c}) = \mathcal{N}(\boldsymbol{x}_t, \sqrt{\bar{\alpha}_t}\boldsymbol{x}_0 + k_t r_\phi(\boldsymbol{x}_0, \boldsymbol{c}, t), (1-\bar{\alpha}_t)\boldsymbol{I}), \tag{3}$$

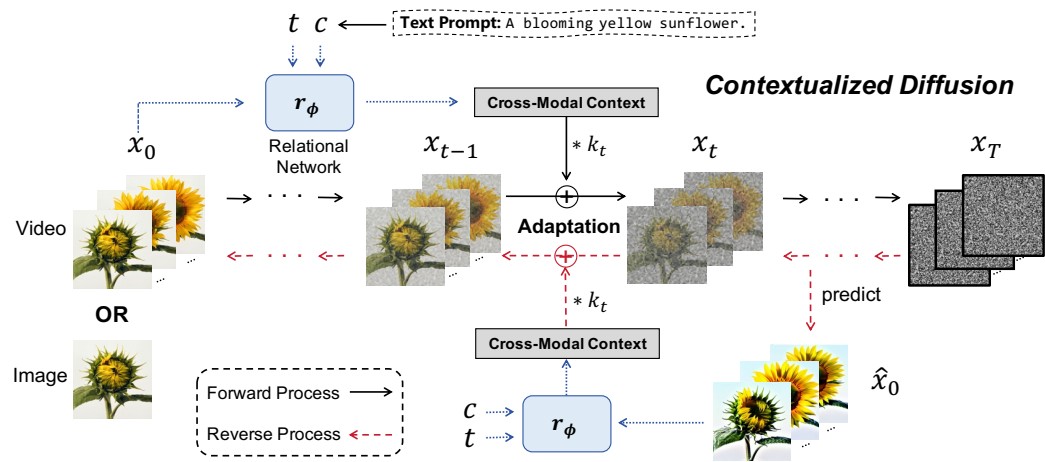

Figure 2: Illustration of our CONTEXTDIFF.

where scalar $k_t$ control the magnitude of the bias term, and we set the $k_t$ to $\sqrt{\bar{\alpha}_t} \cdot (1 - \sqrt{\bar{\alpha}_t})$. $\boldsymbol{r}_\phi(\cdot)$ is the relational network with trainable parameters $\phi$, it takes the visual sample $\boldsymbol{x}_0$ and text condition $\boldsymbol{c}$ as inputs and produces the bias with the same dimension as $\boldsymbol{x}_0$.

Concretely, the forward process is defined as $q_\phi(\boldsymbol{x}_1, \boldsymbol{x}_2, ..., \boldsymbol{x}_T | \boldsymbol{x}_0, \boldsymbol{c}) = \prod_{t=1}^{T} q_\phi(\boldsymbol{x}_t | \boldsymbol{x}_{t-1}, \boldsymbol{x}_0, \boldsymbol{c})$. Given cross-modal context $\boldsymbol{r}_\phi(\boldsymbol{x}_0, \boldsymbol{c}, t)$, the forward transition kernel depends on $\boldsymbol{x}_{t-1}, \boldsymbol{x}_0$, and $\boldsymbol{c}$ :

$$q_\phi(\boldsymbol{x}_t | \boldsymbol{x}_{t-1}, \boldsymbol{x}_0, \boldsymbol{c}) = \mathcal{N}(\sqrt{\alpha_t}\boldsymbol{x}_{t-1} + k_t \boldsymbol{r}_\phi(\boldsymbol{x}_0, \boldsymbol{c}, t) - \sqrt{\alpha_t} k_{t-1} \boldsymbol{r}_\phi(\boldsymbol{x}_0, \boldsymbol{c}, t-1), \beta_t \boldsymbol{I}), \quad (4)$$

where $\beta_t = 1 - \alpha_t$. This transition kernel gives marginal distribution as Equation (3) (proof in Appendix C.1). At each timestep $t$, we add a noise that explicitly biased by the cross-modal context. With Equation (3) and Equation (4), we can derive the posterior distribution of the forward process for $t > 1$ (proof in Appendix C.1):

$$q_\phi(\boldsymbol{x}_{t-1} | \boldsymbol{x}_t, \boldsymbol{x}_0, \boldsymbol{c}) = \mathcal{N}(\frac{\sqrt{\bar{\alpha}_{t-1}}\beta_t}{1-\bar{\alpha}_t}\boldsymbol{x}_0 + \frac{\sqrt{\alpha_t}(1-\bar{\alpha}_{t-1})}{1-\bar{\alpha}_t}(\boldsymbol{x}_t - \boldsymbol{b}_t(\boldsymbol{x}_0, \boldsymbol{c})) + \boldsymbol{b}_{t-1}(\boldsymbol{x}_0, \boldsymbol{c}), \frac{(1-\bar{\alpha}_{t-1})\beta_t}{1-\bar{\alpha}_t}\boldsymbol{I}),$$
$$(5)$$

where $\boldsymbol{b}_t(\boldsymbol{x}_0, \boldsymbol{c})$ is an abbreviation form of $k_t \boldsymbol{r}_\phi(\boldsymbol{x}_0, \boldsymbol{c}, t)$, and we use it for simplicity. With Equation (5), we can simplify the training objective which will be described latter. In this way, we contextualize the entire diffusion process with a context-aware trajectory adapter. In CONTEXTDIFF, we also utilize our context-aware context to adapt the reverse process of diffusion models, which encourages to align with the adapted forward process, and facilitates the precise expression of textual semantics in visual sampling process.

## 4.2 ADAPTING REVERSE PROCESS

We aim to learn a contextualized reverse process $\{p_\theta(\boldsymbol{x}_{t-1} | \boldsymbol{x}_t, \boldsymbol{c})\}_{t=1}^{T}$ , which minimizes a variational upper bound of the negative log likelihood, as in Equation (1). $p_\theta(\boldsymbol{x}_{t-1} | \boldsymbol{x}_t, \boldsymbol{c})$ is gaussian kernel with learnable mean and pre-defined variance. Allowing the forward transition kernel to depend on $\boldsymbol{x}_0$ and $\boldsymbol{c}$, the objective function $\mathcal{L}_{\theta,\phi}$ of our CONTEXTDIFF can be formulated as (proof in Appendix C.2):

$$\mathcal{L}_{\theta,\phi} = \mathbb{E}_{q_\phi(\boldsymbol{x}_{1:T}|\boldsymbol{x}_0,\boldsymbol{c})} \Bigg[ D_{\mathrm{KL}}(q_\phi(\boldsymbol{x}_T | \boldsymbol{x}_0, \boldsymbol{c}) \| p(\boldsymbol{x}_T | \boldsymbol{c})) - \log p_\theta(\boldsymbol{x}_0 | \boldsymbol{x}_1, \boldsymbol{c})$$
$$+ \sum_{t>1} D_{\mathrm{KL}}(q_\phi(\boldsymbol{x}_{t-1} | \boldsymbol{x}_t, \boldsymbol{x}_0, \boldsymbol{c}) \| p_\theta(\boldsymbol{x}_{t-1} | \boldsymbol{x}_t, \boldsymbol{c})) \Bigg], \qquad (6)$$

where $\theta$ denotes the learnable parameters of denoising network in reverse process. Equation (6) uses KL divergence to directly compare $p_\theta(\boldsymbol{x}_{t-1} | \boldsymbol{x}_t, \boldsymbol{c})$ against the adapted forward process posteriors, which are tractable when conditioned on $\boldsymbol{x}_0$ and $\boldsymbol{c}$. If $\boldsymbol{r}_\phi$ is identically zero, the objective can be viewed as the original DDPMs. Thus CONTEXTDIFF is theoretically capable of achieving better likelihood compared to original DDPMs.

Kindly note that optimizing $\mathcal{L}_t = E_{q_\phi} D_{\mathrm{KL}}(q_\phi(\boldsymbol{x}_{t-1} | \boldsymbol{x}_t, \boldsymbol{x}_0, \boldsymbol{c}) \| p_\theta(\boldsymbol{x}_{t-1} | \boldsymbol{x}_t, \boldsymbol{c}))$ is equivalent to matching the means for $q_\phi(\boldsymbol{x}_{t-1} | \boldsymbol{x}_t, \boldsymbol{x}_0, \boldsymbol{c})$ and $p_\theta(\boldsymbol{x}_{t-1} | \boldsymbol{x}_t, \boldsymbol{c})$, as they are gaussian distributions

with the same variance. According to Equation (5), directly matching the means requires to parameterize a neural network $\boldsymbol{\mu}_\theta$ that not only predicting $\boldsymbol{x}_0$, but also matching the complex cross-modal context information in the forward process, i.e.,

$$\mathcal{L}_{\theta,\phi,t} = \left\| \boldsymbol{\mu}_\theta(\boldsymbol{x}_t, c, t) - \frac{\sqrt{\bar{\alpha}_{t-1}}\beta_t}{1-\bar{\alpha}_t}\boldsymbol{x}_0 - \frac{\sqrt{\bar{\alpha}_t}(1-\bar{\alpha}_{t-1})}{1-\bar{\alpha}_t}(\boldsymbol{x}_t - \boldsymbol{b}_t(x_0, \boldsymbol{c})) - \boldsymbol{b}_{t-1}(x_0, \boldsymbol{c}) \right\|_2^2 \quad (7)$$

**Simplified Training Objective** Directly optimizing this objective is inefficient in practice because it needs to compute the bias twice at each timestep. To simplify the training process, we employ a denoising network $\boldsymbol{f}_\theta(\boldsymbol{x}_t, \boldsymbol{c}, t)$ to directly predict $\boldsymbol{x}_0$ from $\boldsymbol{x}_t$ at each time step t, and insert the predicted $\hat{\boldsymbol{x}}_0$ in Equation (5), i.e., $p_{\theta,\phi}(\boldsymbol{x}_{t-1}|\boldsymbol{x}_t, \boldsymbol{c}) = q_\phi(\boldsymbol{x}_{t-1}|\boldsymbol{x}_t, \hat{\boldsymbol{x}}_0, \boldsymbol{c})$. Under mild condition, we can derive that the reconstruction objective $\mathbb{E}\|f_\theta - \boldsymbol{x}_0\|_2^2$ is an upper bound of $\mathcal{L}_t$, and thus an upper bound of negative log likehood (proof in Appendix C.2). Our simplified training objective is:

$$\mathcal{L}_{\theta,\phi} = \sum_{t=1}^{T} \lambda_t \mathbb{E}_{\boldsymbol{x}_0,\boldsymbol{x}_t} \|\boldsymbol{f}_\theta(\boldsymbol{x}_{t,\phi}, \boldsymbol{c}, t) - \boldsymbol{x}_0\|_2^2, \quad (8)$$

where $\lambda_t$ is a weighting scalar. We set $k_T = 0$ and there is no learnable parameters in $D_{\mathrm{KL}}(q(\boldsymbol{x}_T|\boldsymbol{x}_0, \boldsymbol{c})\|p(\boldsymbol{x}_T|\boldsymbol{c}))$, which can be ignored. To adapt the reverse process at each timestep, we can efficiently sample a noisy sample $\boldsymbol{x}_t$ according to Equation (3) using re-parameterization trick, which has included parameterized cross-modal context $k_t\boldsymbol{r}_\phi(\boldsymbol{x}_0, \boldsymbol{c}, t)$, and then passes $\boldsymbol{x}_t$ into the denoising network. The gradients will be propagated to $\boldsymbol{r}_\phi$ from the denoising network, and our context-aware adapter and denoising network are jointly optimized in training.

**Context-Aware Sampling** During sampling, we use the denoising network to predict $\hat{\boldsymbol{x}}_0$, and the predicted context-aware adaptation $\boldsymbol{r}_\phi(\hat{\boldsymbol{x}}_0, \boldsymbol{c}, t)$ is then used to contextualize the sampling trajectory. Hence the gaussian kernel $p_\theta(\boldsymbol{x}_{t-1}|\boldsymbol{x}_t, \boldsymbol{c})$ has mean:

$$\frac{\sqrt{\bar{\alpha}_{t-1}}\beta_t}{1-\bar{\alpha}_t}\hat{\boldsymbol{x}}_0 + \frac{\sqrt{\bar{\alpha}_t}(1-\bar{\alpha}_{t-1})}{1-\bar{\alpha}_t}(\boldsymbol{x}_t - \boldsymbol{b}_t(\hat{\boldsymbol{x}}_0, \boldsymbol{c})) + \boldsymbol{b}_{t-1}(\hat{\boldsymbol{x}}_0, \boldsymbol{c}), \quad (9)$$

where $\boldsymbol{b}_t(\hat{\boldsymbol{x}}_0, \boldsymbol{c}) = k_t\boldsymbol{r}_\phi(\hat{\boldsymbol{x}}_0, c, t)$, and variance $\frac{(1-\bar{\alpha}_{t-1})\beta_t}{1-\bar{\alpha}_t}\boldsymbol{I}$. In this way, our CONTEXTDIFF can effectively adapt sampling process with cross-modal context, which is more informative and contextual guided compared to traditional text guidance (Rombach et al., 2022; Saharia et al., 2022b). Next, we will introduce how to generalize our contextualized diffusion to DDIMs for fast sampling.

## 4.3 GENERALIZING CONTEXTUALIZED DIFFUSION TO DDIMS

DDIMs (Song et al., 2020a) accelerate the reverse process of pretrained DDPMs, which are also faced with the inconsistency problem that exists in DDPMs. Therefore, we address this problem by generalizing our contextualized diffusion to DDIMs. Specifically, we define a posterior distribution $q_\phi(\boldsymbol{x}_{t-1}|\boldsymbol{x}_t, \boldsymbol{x}_0, \boldsymbol{c})$ for each timestep, thus the forward diffusion process has the desired distribution:

$$q_\phi(\boldsymbol{x}_t|\boldsymbol{x}_0, \boldsymbol{c}) = \mathcal{N}(\boldsymbol{x}_t, \sqrt{\bar{\alpha}_t}\boldsymbol{x}_0 + \boldsymbol{b}_t(\boldsymbol{x}_0, \boldsymbol{c}), (1-\bar{\alpha}_t)\boldsymbol{I}), \quad (10)$$

If the posterior distribution is defined as:

$$q(\boldsymbol{x}_{t-1}|\boldsymbol{x}_t, \boldsymbol{x}_0, \boldsymbol{c}) = \mathcal{N}(\sqrt{\bar{\alpha}_{t-1}}\boldsymbol{x}_0 + \sqrt{1-\bar{\alpha}_{t-1}-\sigma_t^2} * \frac{\boldsymbol{x}_t - \sqrt{\bar{\alpha}_t}\boldsymbol{x}_0}{\sqrt{1-\bar{\alpha}_t}}, \sigma_t^2\boldsymbol{I}), \quad (11)$$

then the mean of $q_\phi(\boldsymbol{x}_{t-1}|\boldsymbol{x}_0, \boldsymbol{c})$ is (proof in Appendix C.1):

$$\sqrt{\bar{\alpha}_{t-1}}\boldsymbol{x}_0 + \boldsymbol{b}_t(\boldsymbol{x}_0, \boldsymbol{c}) * \frac{\sqrt{1-\bar{\alpha}_{t-1}-\sigma_t^2}}{\sqrt{1-\bar{\alpha}_t}} \quad (12)$$

To match the forward diffusion, we need to replace the adaptation $k_t\boldsymbol{r}_\phi(\boldsymbol{x}_0, \boldsymbol{c}, t) * \frac{\sqrt{1-\bar{\alpha}_{t-1}-\sigma_t^2}}{\sqrt{1-\bar{\alpha}_t}}$ with $k_{t-1}\boldsymbol{r}_\phi(\boldsymbol{x}_0, \boldsymbol{c}, t-1)$. Given $\sigma_t^2 = 0$, the sampling process becomes deterministic:

$$\tilde{\boldsymbol{x}}_{t-1} = \sqrt{\bar{\alpha}_{t-1}}\hat{\boldsymbol{x}}_0 + \sqrt{1-\bar{\alpha}_{t-1}} * \frac{\boldsymbol{x}_t - \sqrt{\bar{\alpha}_t}\hat{\boldsymbol{x}}_0}{\sqrt{1-\bar{\alpha}_t}}$$

$$\boldsymbol{x}_{t-1} = \tilde{\boldsymbol{x}}_{t-1} - \boldsymbol{b}_t(\hat{\boldsymbol{x}}_0, \boldsymbol{c}) * \frac{\sqrt{1-\bar{\alpha}_{t-1}}}{\sqrt{1-\bar{\alpha}_t}} + \boldsymbol{b}_{t-1}(\hat{\boldsymbol{x}}_0, \boldsymbol{c}). \quad (13)$$

In this way, DDIMs can better convey textual semantics in generated samples when accelerating the sampling of pretrained DDPMs, which will be evaluated in later text-to-video editing task.

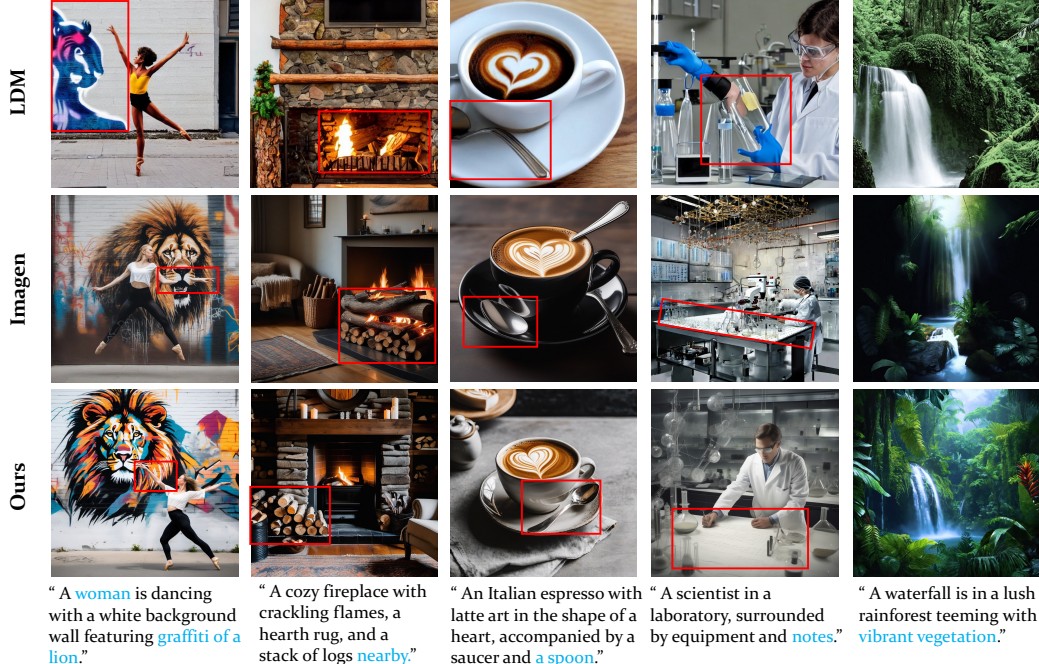

Figure 3: **Qualitative comparison in text-to-image generation.** Our model can better express the semantics of the texts marked in blue. We use red boxes to highlight critical fine-grained parts where LDM and Imagen fail to align with texts.

## 5 EXPERIMENTS

We conduct experiments on two main text-guided visual synthesis tasks: **text-to-image generation (Sec. 5.1) and text-to-video editing (Sec. 5.2)**. We also extend our CONTEXTDIFF to other conditional generation scenarios: **class-to-image and layout-to-image (in Appendix B)**, to demonstrate the generalization ability. For better understanding and explanation of our proposed contextualized diffusion, we further provide some qualitative analysis on FID-CLIP trade-off (Sec. 5.3), model convergence (Sec. 5.3) and heatmap visualization (Appendix A).

### 5.1 TEXT-TO-IMAGE GENERATION

**Datasets and Metrics.** Following Rombach et al. (2022); Saharia et al. (2022b), we use public LAION-400M (Schuhmann et al., 2021), a dataset with CLIP-filtered 400 million image-text pairs for training CONTEXTDIFF. We conduct evaluations with FID and CLIP score (Hessel et al., 2021; Radford et al., 2021), which aim to assess the generation quality and resulting image-text alignment.

**Implementation Details.** For our context-aware adapter, we use text CLIP and image CLIP (Radford et al., 2021) (ViT-B/32) to encode text and image inputs, and adopt multi-head cross attention (Vaswani et al., 2017) to model cross-modal interactions with 8 parallel attention layers. For the diffusion backbone, we mainly follow Imagen (Saharia et al., 2022b) using a $64 \times 64$ base diffusion model (Nichol & Dhariwal, 2021; Saharia et al., 2022a) and a super-resolution diffusion models to upsample a $64 \times 64$ generated image into a $256 \times 256$ image. For $64 \times 64 \rightarrow 256 \times 256$ super-resolution, we use the efficient U-Net model in Imagen for improving memory efficiency. We condition on the entire sequence of text embeddings (Raffel et al., 2020) by adding cross attention (Ramesh et al., 2022) over the text embeddings at multiple resolutions. More details about the hyper-parameters can be found in Appendix E.

**Quantitative and Qualitative Results** Following previous works (Rombach et al., 2022; Ramesh et al., 2022; Saharia et al., 2022b), we make quantitative evaluations CONTEXTDIFF on the MS-COCO dataset using zero-shot FID score, which measures the quality and diversity of generated images. Similar to Rombach et al. (2022); Ramesh et al. (2022); Saharia et al. (2022b), 30,000 images are randomly selected from the validation set for evaluation. As demonstrated in Tab. 1, our CONTEXTDIFF achieves a new state-of-the-art performance on text-to-image generation task with

Table 1: **Quantitative results in text-to-image generation** with FID score on MS-COCO dataset for 256 × 256 image resolution.

| Approach | Model Type | FID-30K | Zero-shot FID-30K |
|---|---|---|---|
| DF-GAN (Tao et al., 2022) | GAN | 21.42 | - |
| DM-GAN + CL (Ye et al., 2021) | GAN | 20.79 | - |
| LAFITE (Zhou et al., 2022) | GAN | 8.12 | - |
| Make-A-Scene (Gafni et al., 2022) | Autoregressive | 7.55 | - |
| DALL-E (Ramesh et al., 2021) | Autoregressive | - | 17.89 |
| Stable Diffusion (Rombach et al., 2022) | Continuous Diffusion | - | 12.63 |
| GLIDE (Nichol et al., 2022b) | Continuous Diffusion | - | 12.24 |
| DALL-E 2 (Ramesh et al., 2022) | Continuous Diffusion | - | 10.39 |
| Improved VQ-Diffusion (Tang et al., 2022) | Discrete Diffusion | - | 8.44 |
| Simple Diffusion (Hoogeboom et al., 2023) | Continuous Diffusion | - | 8.32 |
| Imagen (Saharia et al., 2022b) | Continuous Diffusion | - | 7.27 |
| Parti (Yu et al., 2022) | Autoregressive | - | 7.23 |
| Muse (Chang et al., 2023) | Non-Autoregressive | - | 7.88 |
| eDiff-I (Balaji et al., 2022) | Continuous Diffusion | - | 6.95 |
| ERNIE-ViLG 2.0 (Feng et al., 2023) | Continuous Diffusion | - | 6.75 |
| RAPHAEL (Xue et al., 2023) | Continuous Diffusion | - | 6.61 |
| **CONTEXTDIFF** | Continuous Diffusion | - | **6.48** |

Table 2: **Quantitative results in text-to-video editing.** Text. and Temp. denote CLIP-text and CLIP-temp, respectively. User study shows the preference rate of CONTEXTDIFF against baselines via human evaluation.

| Method | Metric | | User Study | |
|---|---|---|---|---|
| | Text.↑ | Temp.↑ | Text. (%)↑ | Temp.(%)↑ |
| Tune-A-Video (Wu et al., 2022) | 0.260 | 0.934 | 91 | 84 |
| FateZero (Qi et al., 2023) | 0.252 | 0.954 | 84 | 75 |
| ControlVideo (Zhao et al., 2023) | 0.258 | 0.961 | 81 | 73 |
| **CONTEXTDIFF** | **0.274** | **0.970** | - | - |

6.48 zero-shot FID score, outperforming previous dominant diffusion models such as Stable Diffusion (Rombach et al., 2022), DALL-E 2 (Ramesh et al., 2022), and Imagen (Saharia et al., 2022b). We also make qualitative comparisons in Figure 3, and find that our CONTEXTDIFF can achieve more precise semantic alignment between text prompt and generated image than previous methods, demonstrating the effectiveness of incorporating cross-modal context into diffusion models. We visualize more qualitative results in Appendix F.1.

## 5.2 TEXT-TO-VIDEO EDITING

**Datasets and Metrics** To demonstrate the strength of our CONTEXTDIFF for text-to-video edting, we use 42 representative videos taken from DAVIS dataset (Pont-Tuset et al., 2017) and other in-the-wild videos following previous works (Wu et al., 2022; Qi et al., 2023; Bar-Tal et al., 2022; Esser et al., 2023). These videos cover a range of categories including animals, vehicles, and humans. To obtain video footage, we use BLIP-2 (Li et al., 2023) for automated captions. We also use their designed prompts for each video, including object editing, background changes, and style transfers. To measure textual alignment, we compute average CLIP score between all frames of output videos and corresponding edited prompts. For temporal consistency, we compute CLIP (Radford et al., 2021) image embeddings on all frames of output videos and report the average cosine similarity between all pairs of video frames. Moreover, We perform user study to quantify text alignment, and temporal consistency by pairwise comparisons between the baselines and our CONTEXTDIFF. A total of 10 subjects participated in this user study. Taking text alignment as an example, given a source video, the participants are instructed to select which edited video is more aligned with the text prompt in the pairwise comparisons between the baselines and CONTEXTDIFF.

**Implementation Details** In order to reproduce the baselines of Tune-A-Video (Wu et al., 2022), FateZero (Qi et al., 2023), and ControlVideo (Zhao et al., 2023), we use their official repositories

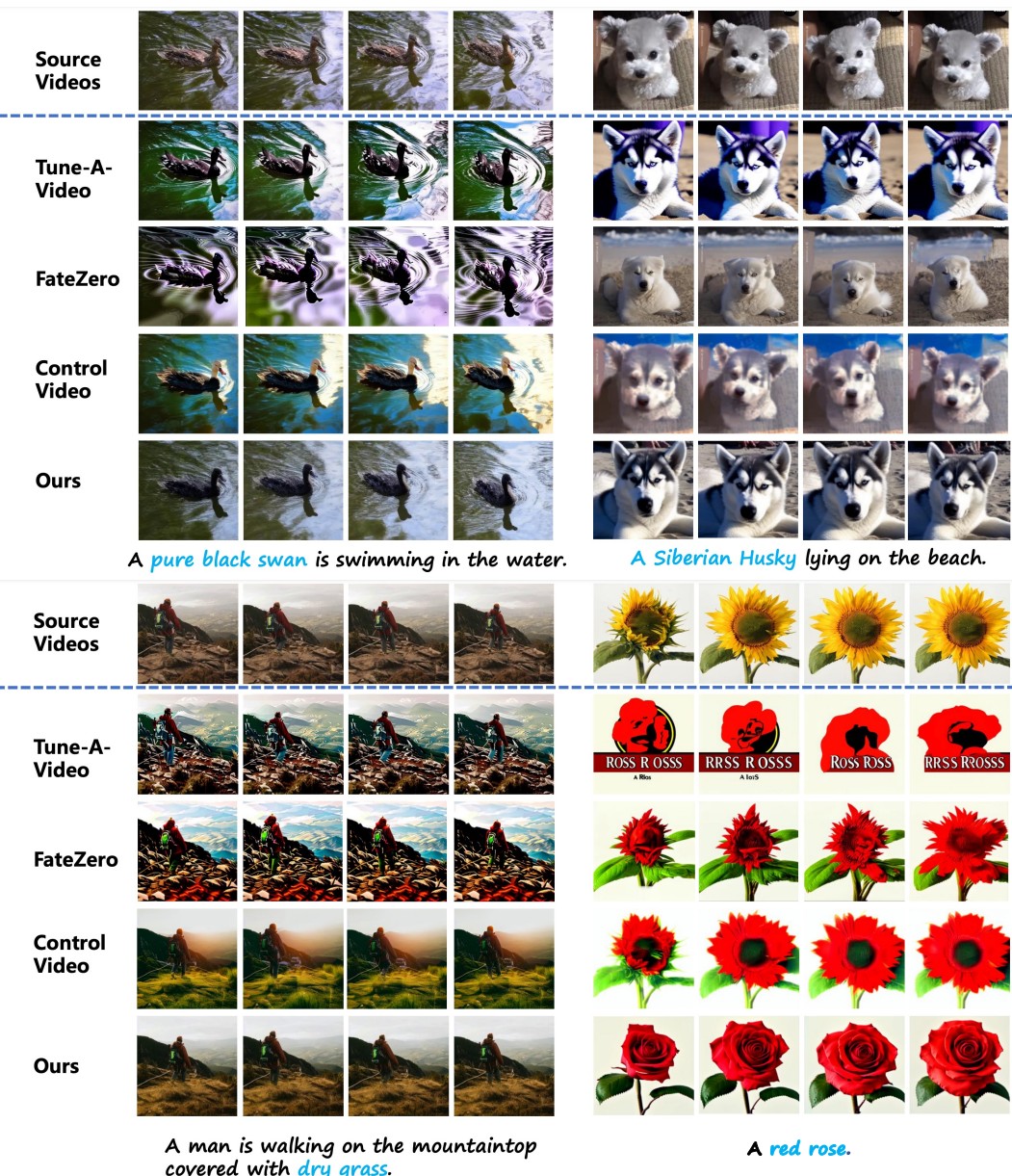

Figure 4: **Qualitative comparison in text-to-video editing**, edited text prompt is denoted in color. Our CONTEXTDIFF achieves best semantic alignment, image fidelity, and editing quality.

for one-shot video tuning. Following FateZero, we use the trained Stable Diffusion v1.4 (Rombach et al., 2022) as the base text-to-image diffusion model, and fuse the attention maps in DDIM inversion (Song et al., 2020a) and sampling processes for retaining both structural and motion information. We fuse the attentions in the interval of $t \in [0.5 \times T, T]$ of the DDIM step with total timestep T = 20. For context-aware adapter, we use the same encoders and cross attention as in text-to-image generation. We additionally incorporate spatio-temporal attention, which includes spatial self-attention and temporal causal attention, into our context-aware adapter for capturing spatio-temporal consistency. For each source video, we tune our adapter using source text prompt for learning both context-aware structural and motion information, and use the learned adapter to conduct video editing with edited text prompt. Details about the hyper-parameters are in Appendix E.

**Quantitative and Qualitative Results** We report our quantitative and qualitative results in Tab. 2 and Figure 4. Extensive results demonstrate that CONTEXTDIFF substantially outperforms all these baselines in both textual alignment and temporal consistency. Notably, in the textual alignment in user study, we outperform the baseline by a significant margin (over 80%), demonstrating the su-

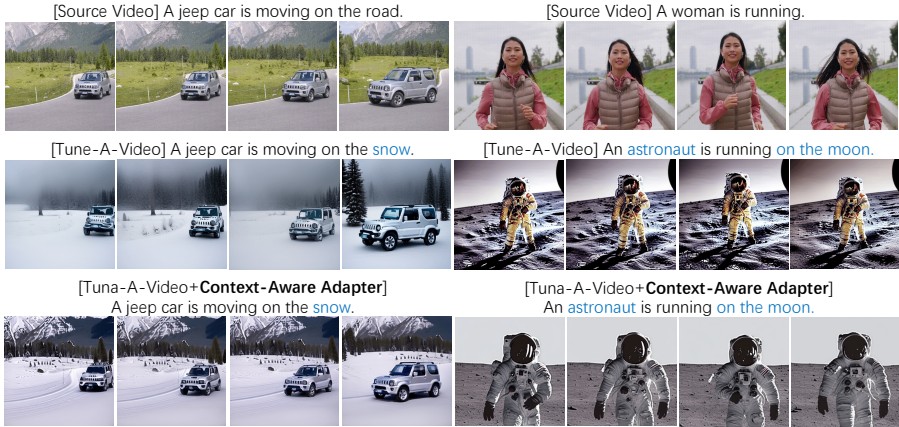

Figure 5: Generalizing our context-aware adapter to Tune-A-Video (Wu et al., 2022).

perior cross-modal understanding of our contextualized diffusion. In qualitative comparisons, we observe that CONTEXTDIFF not only achieves better semantic alignment, but also preserves the structure information in source video. Besides, the context-aware adapter in our contextualized diffusion can be generalized to previous methods, which substantially improves the generation quality as in Figure 5. More results demonstrating our generalization ability can be found in Appendix F.2.

### 5.3 ABLATION STUDY

**Guidance Scale *vs*. FID**   Given the significance of classifier-free guidance weight in controlling image quality and text alignment, in Figure 6, we conduct ablation study on the trade-off between CLIP and FID scores across a range of guidance weights, specifically 1.5, 3.0, 4.5, 6.0, 7.5, and 9.0. The results indicate that our context-aware adapter contribute effectively. At the same guidance weight, our context-aware adapter considerably and consistently reduces the FID, resulting in a significant improvement in image quality.

**Training Convergence**   We evaluate CONTEXTDIFF regarding our contribution to the model convergence. The comparison in Figure 7 demonstrates that our context-aware adapter can significantly accelerate the training convergence and improve the semantic alignment between text and generated video. This observation also reveals the generalization ability of our contextualized diffusion.

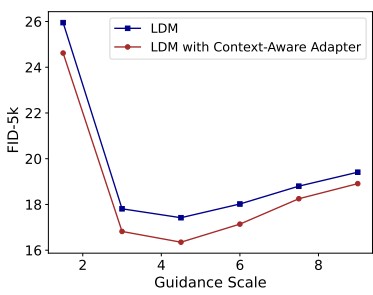

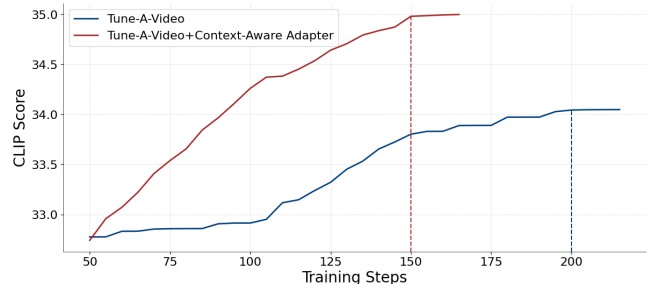

Figure 6: The trade-off between FID and CLIP scores for LDM and LDM with our context-aware adapter.

Figure 7: The comparison of model convergence between Tune-A-Video and Tune-A-Video + our context-aware adapter.

### 6 CONCLUSION

In this paper, we propose a novel and general conditional diffusion model (CONTEXTDIFF) by propagating cross-modal context to all timesteps in both diffusion and reverse processes, and adapt their trajectories for facilitating the model capacity of cross-modal synthesis. We generalize our contextualized trajectory adapter to DDPMs and DDIMs with theoretical derivation, and consistently achieve state-of-the-art performance in two challenging tasks: text-to-image generation, and text-to-video editing. Extensive quantitative and qualitative results on the two tasks demonstrate the effectiveness and superiority of our proposed cross-modal contextualized diffusion models.

## ACKNOWLEDGEMENT

This work was supported by the National Natural Science Foundation of China (No.U23B2048 and U22B2037).

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

## A    VISUAL ANALYSIS ON CONTEXT AWARENESS OF OUR MODEL

We conduct visual analysis to investigate how our context-aware adapter works in text-guided visual synthesis. As illustrated in Figure 8 and Figure 9, we visualize the heatmaps of text-image cross-attention module in the sampling process of each frame image. We find that our context-aware adapter can enable the model to better focus on the fine-grained semantics in text prompt (Lei et al., 2022; Wang et al., 2023a) and sufficiently convey them in final generation results. Because incorporating textual information into diffusion process of the image benefits the cross-modal understanding for image diffusion models.

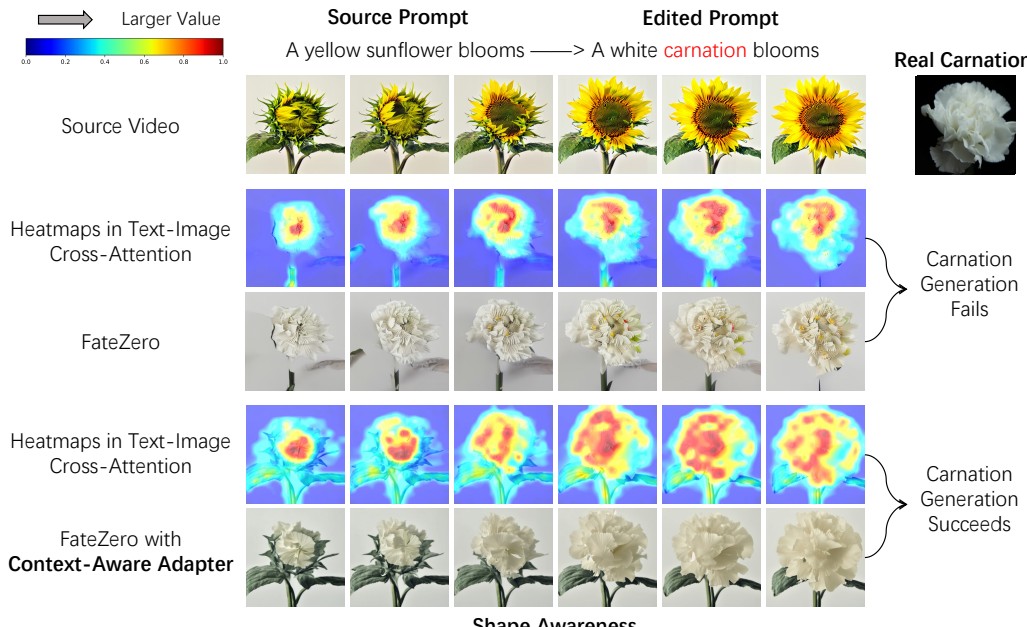

Figure 8: Our context-aware adapter improves the shape awareness of diffusion models in text-guided video editing.

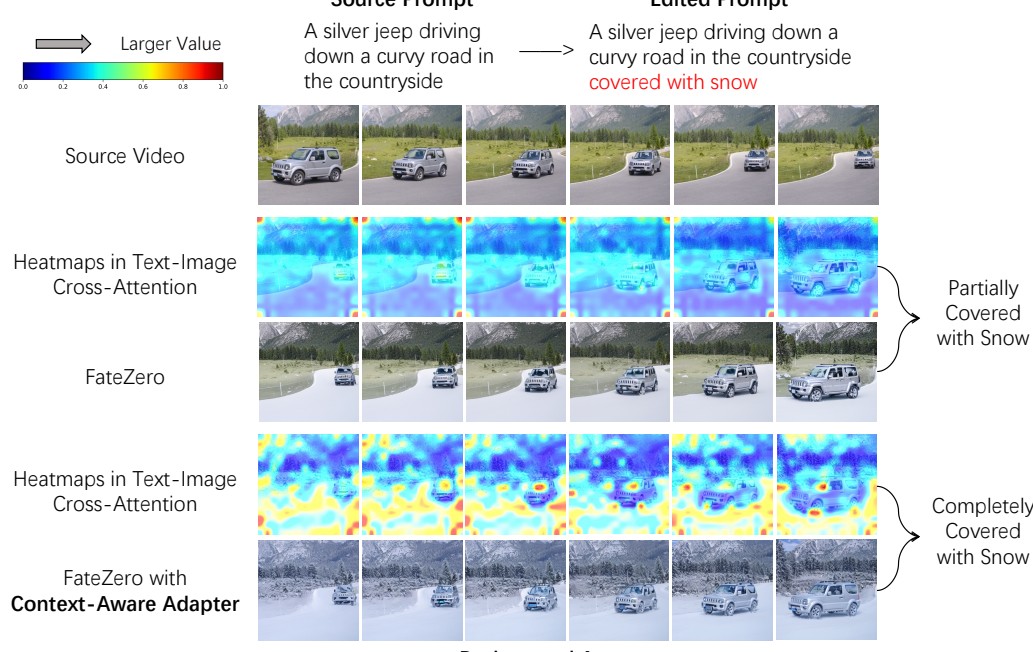

Figure 9: Our context-aware adapter can improve the background awareness of diffusion models in text-guided video editing.

# B  GENERALIZING TO CLASS AND LAYOUT CONDITIONAL GENERATION TASKS

We generalize our context-aware adapter into class and layout conditional generation tasks. We replace the text encoder in original adapter with ResNet blocks for embedding classes or layouts, and keep the original image encoder and cross-attention module for obtaining cross-modal context information. We put both quantitative and qualitative results in Tabs. 3 and 4 and Figures 10 and 11. From the results, we conclude that our context-aware adapter can benefit the conditional diffusion models with different condition modalities and enable more realistic and precise generation consistent with input conditions, demonstrating the satisfying generalization ability of our method.

Table 3: Performance comparison in class-to-image generation on ImageNet 256×256.

| Method | FID ↓ | IS ↑ | Precision ↑ | Recall ↑ |
| --- | --- | --- | --- | --- |
| BigGAN (Brock et al., 2018) | 6.95 | 203.63 | 0.87 | 0.28 |
| ADM-G (Dhariwal & Nichol, 2021) | 4.59 | 186.70 | 0.82 | 0.52 |
| LDM (Rombach et al., 2022) | 3.60 | 247.67 | 0.87 | 0.48 |
| LDM+**Context-Aware Adapter** | **2.97** | **273.04** | **0.89** | **0.55** |

Table 4: FID performance comparison in layout-to-image generation on MS-COCO 256×256.

| Method | FID↓ |
| --- | --- |
| VQGAN+T (Jahn et al., 2021) | 56.58 |
| Frido (Fan et al., 2023) | 37.14 |
| LDM (Rombach et al., 2022) | 40.91 |
| LDM+**Context-Aware Adapter** | **34.58** |

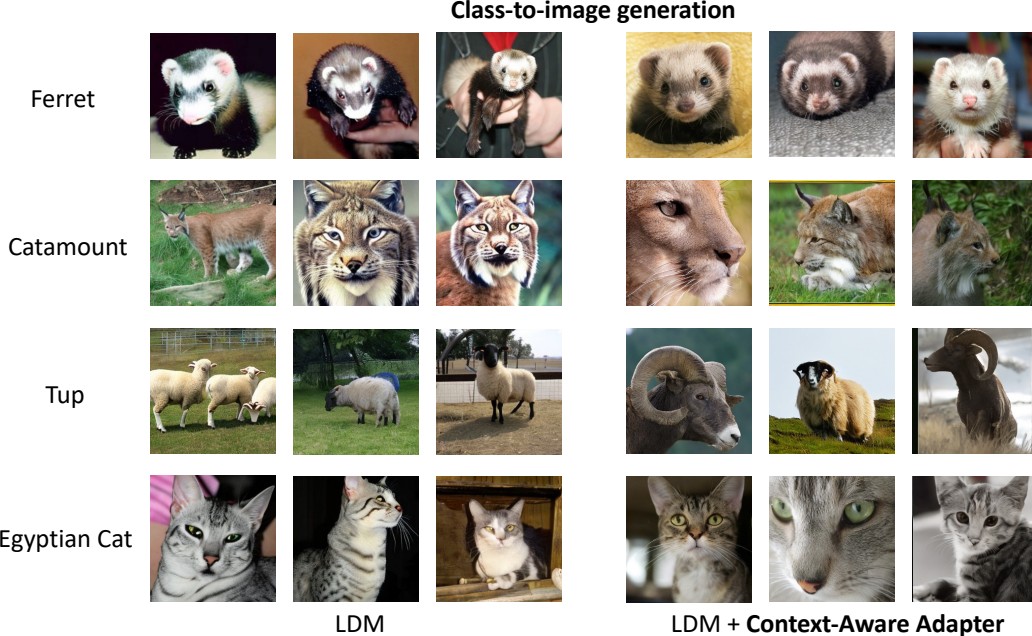

Figure 10: Qualitative results in class-to-image generation on ImageNet 256×256. Our context-aware adapter improves the generation quality of LDM.

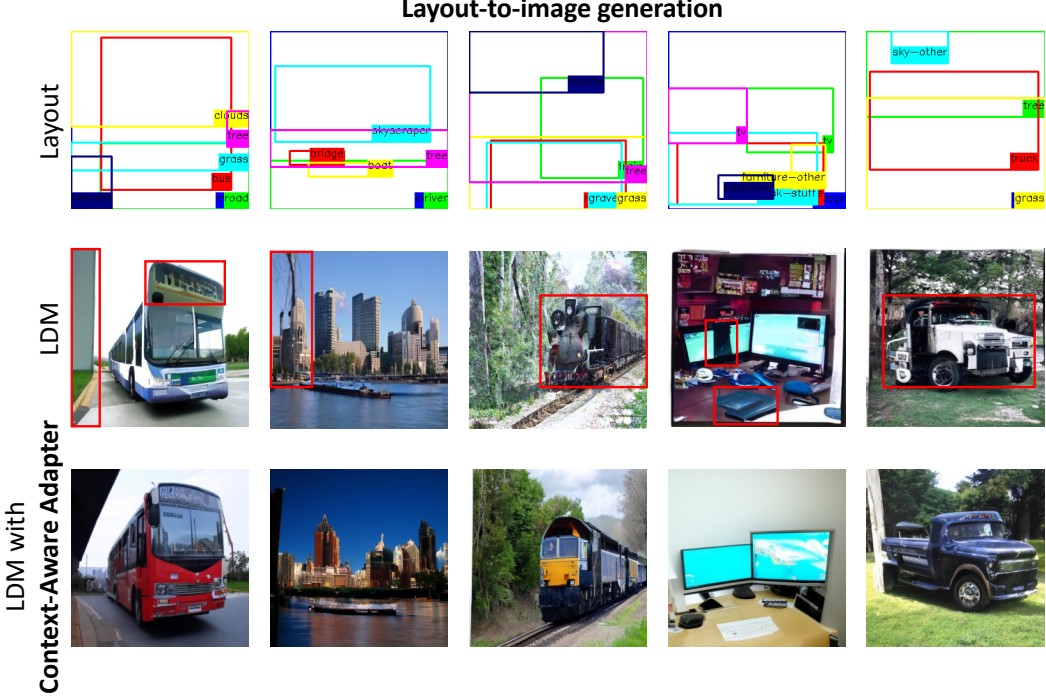

Figure 11: Qualitative results in layout-to-image generation on MS-COCO 256×256. Our context-aware adapter improves both fidelity and precision of the generation results of LDM. We use red boxes to highlight critical fine-grained parts where original LDM fails to align with conditional layout. Our method substantially improves both quality and precision of the generation results.

## C  THEORETICAL DERIVATIONS

### C.1  THE DISTRIBUTIONS IN THE FORWARD PROCESS

First, we derive the explicit expressions for $q(\boldsymbol{x}_t|\boldsymbol{x}_{t-1}, \boldsymbol{x}_0, \boldsymbol{c})$ and $q(\boldsymbol{x}_{t-1}|\boldsymbol{x}_t, \boldsymbol{x}_0, \boldsymbol{c})$, based on our cross-modal contextualized diffusion defined by Equation (3).

**lemma 1** *For the forward process $q(\boldsymbol{x}_1, \boldsymbol{x}_2, ..., \boldsymbol{x}_T|\boldsymbol{x}_0, \boldsymbol{c}) = \prod_{t=1}^{T} q(\boldsymbol{x}_t|\boldsymbol{x}_{t-1}, \boldsymbol{x}_0, \boldsymbol{c})$, if the transition kernel $q(\boldsymbol{x}_t|\boldsymbol{x}_{t-1}, \boldsymbol{x}_0, \boldsymbol{c})$ is defined as Equation (4), then the conditional distribution $q(\boldsymbol{x}_t|\boldsymbol{x}_0, \boldsymbol{c})$ has the desired distribution as Equation (3), i.e., $\mathcal{N}(\boldsymbol{x}_t, \sqrt{\bar{\alpha}_t}\boldsymbol{x}_0 + k_t \boldsymbol{r}_\phi(\boldsymbol{x}_0, \boldsymbol{c}, t), (1 - \bar{\alpha}_t)\boldsymbol{I})$.*

**Proof 1** *We prove the lemma by induction. Suppose at time $t$, we have $q(\boldsymbol{x}_t|\boldsymbol{x}_{t-1}, \boldsymbol{x}_0, \boldsymbol{c})$ and $q(\boldsymbol{x}_{t-1}|\boldsymbol{x}_0, \boldsymbol{c})$ admit the desired distributions as in Equations (3) and (4), respectively, then we need to prove that $q(\boldsymbol{x}_t|\boldsymbol{x}_0, \boldsymbol{c}) = \mathcal{N}(\boldsymbol{x}_t, \sqrt{\bar{\alpha}_t}\boldsymbol{x}_0 + k_t \boldsymbol{r}_\phi(\boldsymbol{x}_0, \boldsymbol{c}, t), (1 - \bar{\alpha}_t)\boldsymbol{I})$. We can re-write he conditional distributions of $\boldsymbol{x}_t$ given $(\boldsymbol{x}_{t-1}, \boldsymbol{x}_0, \boldsymbol{c})$ and $\boldsymbol{x}_{t-1}$ given $(\boldsymbol{x}_0, \boldsymbol{c})$ with the following equations:*

$$\boldsymbol{x}_t = \sqrt{\alpha_t}\boldsymbol{x}_{t-1} + k_t \boldsymbol{r}_\phi(\boldsymbol{x}_0, \boldsymbol{c}, t) - \sqrt{\alpha_t} k_{t-1} \boldsymbol{r}_\phi(\boldsymbol{x}_0, \boldsymbol{c}, t-1) + \sqrt{\beta_t}\epsilon_1, \tag{14}$$

$$\boldsymbol{x}_{t-1} = \sqrt{\bar{\alpha}_{t-1}}\boldsymbol{x}_0 + k_{t-1} \boldsymbol{r}_\phi(\boldsymbol{x}_0, \boldsymbol{c}, t-1) + \sqrt{1 - \bar{\alpha}_{t-1}}\epsilon_2, \tag{15}$$

*where $\epsilon_1, \epsilon_2$ are two independent standard gaussian random variables. Replacing $\boldsymbol{x}_{t-1}$ in Equation (14) with Equation (15), we have:*

$$\begin{aligned}
\boldsymbol{x}_t &= \sqrt{\bar{\alpha}_t}\boldsymbol{x}_0 + k_t \boldsymbol{r}_\phi(\boldsymbol{x}_0, \boldsymbol{c}, t) \\
&\quad + \sqrt{\alpha_t} k_{t-1} \boldsymbol{r}_\phi(\boldsymbol{x}_0, \boldsymbol{c}, t-1) - \sqrt{\alpha_t} k_{t-1} \boldsymbol{r}_\phi(\boldsymbol{x}_0, \boldsymbol{c}, t-1) \\
&\quad + \sqrt{\beta_t}\epsilon_1 + \sqrt{\alpha_t * (1 - \bar{\alpha}_{t-1})} * \epsilon_2 \\
&= \sqrt{\bar{\alpha}_t}\boldsymbol{x}_0 + k_{t-1} \boldsymbol{r}_\phi(\boldsymbol{x}_0, \boldsymbol{c}, t-1) + \sqrt{\beta_t}\epsilon_1 + \sqrt{\alpha_t * (1 - \bar{\alpha}_{t-1})} * \epsilon_2
\end{aligned} \tag{16}$$

As a result, the distribution of $\boldsymbol{x}_t$ given $(\boldsymbol{x}_0, \boldsymbol{c})$ is a gaussian distribution with mean $\sqrt{\bar{\alpha}_t}\boldsymbol{x}_0 + k_t \boldsymbol{r}_\phi(\boldsymbol{x}_0, \boldsymbol{c}, t)$ and variance $\alpha_t * (1 - \bar{\alpha}_{t-1}) + \beta_t = 1 - \bar{\alpha}_t$, which admits the desired distribution.

**Proposition 1** *Suppose the distribution of forward process is defined by Equations (3) and (4), then at each time t, the posterior distribution $q(\boldsymbol{x}_{t-1}|\boldsymbol{x}_t, \boldsymbol{x}_0, \boldsymbol{c})$ is described by Equation (5)*

**Proof 2** *By the Bayes rule, $q(\boldsymbol{x}_{t-1}|\boldsymbol{x}_t, \boldsymbol{x}_0, \boldsymbol{c}) = \frac{q(\boldsymbol{x}_{t-1}|\boldsymbol{x}_0, \boldsymbol{c})q(\boldsymbol{x}_t|\boldsymbol{x}_{t-1}, \boldsymbol{x}_0, \boldsymbol{c})}{q(\boldsymbol{x}_t|\boldsymbol{x}_0, \boldsymbol{c})}$. By Equations (3) and (4), the numerator and denominator are both gaussian , then the posterior distribution is also gaussian and we can proceed to calculate its mean and variance:*

$$q(\boldsymbol{x}_{t-1}|\boldsymbol{x}_t, \boldsymbol{x}_0, \boldsymbol{c}) = \frac{\mathcal{N}(\boldsymbol{x}_{t-1}, \sqrt{\bar{\alpha}_{t-1}}\boldsymbol{x}_0 + \boldsymbol{b}_{t-1}(\boldsymbol{x}_0, \boldsymbol{c}), (1 - \bar{\alpha}_{t-1})\boldsymbol{I})}{\mathcal{N}(\boldsymbol{x}_t, \sqrt{\bar{\alpha}_t}\boldsymbol{x}_0 + \boldsymbol{b}_t(\boldsymbol{x}_0, \boldsymbol{c}), (1 - \bar{\alpha}_t)\boldsymbol{I})} , \quad (17)$$
$$* \mathcal{N}(\boldsymbol{x}_t, \sqrt{\alpha_t}\boldsymbol{x}_{t-1} + \boldsymbol{b}_t(\boldsymbol{x}_0, \boldsymbol{c}) - \sqrt{\alpha_t}\boldsymbol{b}_{t-1}(\boldsymbol{x}_0, \boldsymbol{c}), \beta_t\boldsymbol{I})$$

*where $\boldsymbol{b}_t(\boldsymbol{x}_0, \boldsymbol{c})$ is an abbreviation form of $k_t\boldsymbol{r}_\phi(\boldsymbol{x}_0, \boldsymbol{c}, t)$. Dropping the constants which are unrelated to $\boldsymbol{x}_0, \boldsymbol{x}_t, \boldsymbol{x}_{t-1}$ and $\boldsymbol{c}$, we have:*

$$q(\boldsymbol{x}_{t-1}|\boldsymbol{x}_t, \boldsymbol{x}_0, \boldsymbol{c}) \propto exp\left\{ -\frac{(\boldsymbol{x}_{t-1} - \sqrt{\bar{\alpha}_{t-1}}\boldsymbol{x}_0 - \boldsymbol{b}_{t-1}(\boldsymbol{x}_0, \boldsymbol{c}))^2}{2(1 - \bar{\alpha}_{t-1})} + \frac{(\boldsymbol{x}_t - \sqrt{\bar{\alpha}_t}\boldsymbol{x}_0 - \boldsymbol{b}_t(\boldsymbol{x}_0, \boldsymbol{c}))^2}{2(1 - \bar{\alpha}_t)} \right.$$
$$\left. -\frac{(\boldsymbol{x}_t - \sqrt{\alpha_t}\boldsymbol{x}_{t-1} - \boldsymbol{b}_t(\boldsymbol{x}_0, \boldsymbol{c}) + \sqrt{\alpha_t}\boldsymbol{b}_{t-1}(\boldsymbol{x}_0, \boldsymbol{c}))^2}{2\beta_t} \right\}$$
$$= exp\left\{ C(\boldsymbol{x}_0, \boldsymbol{x}_t, \boldsymbol{c}) - \frac{1}{2}(\frac{1}{1 - \bar{\alpha}_{t-1}} + \frac{\alpha_t}{\beta_t}) * \boldsymbol{x}_{t-1}^2 + \boldsymbol{x}_{t-1}* \right.$$
$$\left. [\frac{(\sqrt{\bar{\alpha}_{t-1}}\boldsymbol{x}_0 + \boldsymbol{b}_{t-1}(\boldsymbol{x}_0, \boldsymbol{c}))}{1 - \bar{\alpha}_{t-1}} + \sqrt{\alpha_t}\frac{(\boldsymbol{x}_t - \boldsymbol{b}_t(\boldsymbol{x}_0, \boldsymbol{c}) + \sqrt{\alpha_t}\boldsymbol{b}_{t-1}(\boldsymbol{x}_0, \boldsymbol{c}))}{\beta_t}] \right\}$$
$$= exp\left\{ C(\boldsymbol{x}_0, \boldsymbol{x}_t, \boldsymbol{c}) - \frac{1}{2}(\frac{1}{1 - \bar{\alpha}_{t-1}} + \frac{\alpha_t}{\beta_t}) * \boldsymbol{x}_{t-1}^2 + \boldsymbol{x}_{t-1}* \right.$$
$$\left. [\frac{(\sqrt{\bar{\alpha}_{t-1}}}{1 - \bar{\alpha}_{t-1}}\boldsymbol{x}_0 + \frac{\sqrt{\alpha_t}}{\beta_t}(\boldsymbol{x}_t - \boldsymbol{b}_t(\boldsymbol{x}_0, \boldsymbol{c})) + (\frac{1}{1 - \bar{\alpha}_{t-1}} + \frac{\alpha_t}{\beta_t}) * \boldsymbol{b}_{t-1}(\boldsymbol{x}_0, \boldsymbol{c})] \right\},$$
$$(18)$$

*where $C(\boldsymbol{x}_0, \boldsymbol{x}_t, \boldsymbol{c})$ is a constant term with respect to $\boldsymbol{x}_{t-1}$. Note that $(\frac{1}{1 - \bar{\alpha}_{t-1}} + \frac{\alpha_t}{\beta_t}) = \frac{1 - \bar{\alpha}_t}{(1 - \bar{\alpha}_{t-1})(1 - \alpha_t)}$, and with some algebraic derivation, we can show that the gaussian distribution $q(\boldsymbol{x}_{t-1}|\boldsymbol{x}_t, \boldsymbol{x}_0, \boldsymbol{c})$ has:*

$$variance : \frac{(1 - \bar{\alpha}_{t-1})(1 - \alpha_t)}{1 - \bar{\alpha}_t}\boldsymbol{I}$$
$$(19)$$
$$mean : \frac{\sqrt{\bar{\alpha}_{t-1}}\beta_t}{1 - \bar{\alpha}_t}\boldsymbol{x}_0 + \frac{\sqrt{\alpha_t}(1 - \bar{\alpha}_{t-1})}{1 - \bar{\alpha}_t}(\boldsymbol{x}_t - \boldsymbol{b}_t(\boldsymbol{x}_0, \boldsymbol{c})) + \boldsymbol{b}_{t-1}(\boldsymbol{x}_0, \boldsymbol{c})$$

Similarly, we can derive the distribution of DDIMs.

**lemma 2** *Suppose that at each time t, the posterior distribution is defined by a gaussin distribution with*

$$Mean : \sqrt{\bar{\alpha}_{t-1}}\boldsymbol{x}_0 + \sqrt{1 - \bar{\alpha}_{t-1} - \sigma_t^2} * \frac{\boldsymbol{x}_t - \sqrt{\bar{\alpha}_t}\boldsymbol{x}_0}{\sqrt{1 - \bar{\alpha}_t}}$$
$$- k_t\boldsymbol{r}_\phi(\boldsymbol{x}_0, \boldsymbol{c}, t) * \frac{\sqrt{1 - \bar{\alpha}_{t-1} - \sigma_t^2}}{\sqrt{1 - \bar{\alpha}_t}} + k_{t-1}\boldsymbol{r}_\phi(\boldsymbol{x}_0, \boldsymbol{c}, t - 1) \quad (20)$$
$$Variance : \sigma_t^2\boldsymbol{I},$$

*then the marginal distribution $q_\phi(\boldsymbol{x}_t|\boldsymbol{x}_0, \boldsymbol{c})$ has the desired distribution as Equation (3)*

**Proof 3** *We prove by induction. Suppose that at time t, posterior and marginal distributions admit the desired distributions, then we need to prove that at time $t-1$, $q_\phi(\boldsymbol{x}_{t-1}|\boldsymbol{x}_0, \boldsymbol{c})$ also has the desired*

*distribution. Rewrite the posterior and marginal distribution as the following:*

$$
\begin{aligned}
\boldsymbol{x}_{t-1} = & \sqrt{\bar{\alpha}_{t-1}}\boldsymbol{x}_0 + \sqrt{1 - \bar{\alpha}_{t-1} - \sigma_t^2} * \frac{\boldsymbol{x}_t - \sqrt{\bar{\alpha}_t}\boldsymbol{x}_0}{\sqrt{1 - \bar{\alpha}_t}} \\
& - k_t \boldsymbol{r}_\phi(\boldsymbol{x}_0, \boldsymbol{c}, t) * \frac{\sqrt{1 - \bar{\alpha}_{t-1} - \sigma_t^2}}{\sqrt{1 - \bar{\alpha}_t}} + k_{t-1}\boldsymbol{r}_\phi(\boldsymbol{x}_0, \boldsymbol{c}, t-1)) + \sigma_t \epsilon_1
\end{aligned}
\tag{21}
$$

$$
\boldsymbol{x}_t = \sqrt{\bar{\alpha}_t}\boldsymbol{x}_0 + k_t \boldsymbol{r}_\phi(\boldsymbol{x}_0, \boldsymbol{c}, t) + \sqrt{1 - \bar{\alpha}_t}\epsilon_2,
\tag{22}
$$

*where $\epsilon_1, \epsilon_2$ are standard gaussian noises. Plugging in $\boldsymbol{x}_t$, we have:*

$$
\begin{aligned}
\boldsymbol{x}_{t-1} = & \sqrt{\bar{\alpha}_{t-1}}\boldsymbol{x}_0 \\
& + k_t \boldsymbol{r}_\phi(\boldsymbol{x}_0, \boldsymbol{c}, t) * \frac{\sqrt{1 - \bar{\alpha}_{t-1} - \sigma_t^2}}{\sqrt{1 - \bar{\alpha}_t}} - k_t \boldsymbol{r}_\phi(\boldsymbol{x}_0, \boldsymbol{c}, t) * \frac{\sqrt{1 - \bar{\alpha}_{t-1} - \sigma_t^2}}{\sqrt{1 - \bar{\alpha}_t}} \\
& + k_{t-1}\boldsymbol{r}_\phi(\boldsymbol{x}_0, \boldsymbol{c}, t-1)) + \sigma_t \epsilon_1 + \sqrt{1 - \bar{\alpha}_{t-1} - \sigma_t^2}\epsilon_2 \\
= & \sqrt{\bar{\alpha}_{t-1}}\boldsymbol{x}_0 + k_{t-1}\boldsymbol{r}_\phi(\boldsymbol{x}_0, \boldsymbol{c}, t-1)) + \sigma_t \epsilon_1 + \sqrt{1 - \bar{\alpha}_{t-1} - \sigma_t^2}\epsilon_2
\end{aligned}
\tag{23}
$$

*Since the variance of $\sigma_t \epsilon_1 + \sqrt{1 - \bar{\alpha}_{t-1} - \sigma_t^2}\epsilon_2$ is $(1 - \bar{\alpha}_{t-1})\boldsymbol{I}$, we have the desired distribution.*

## C.2  UPPER BOUND OF THE LIKELIHOOD

Here we show with our parameterization, the objective function $\mathcal{L}_{\theta,\phi}$ Equation (8) is a upper bound of the negative log likelihood of the data distribution.

**lemma 3** *Based on the non-Markovian forward process $q(\boldsymbol{x}_1, \boldsymbol{x}_2, ..., \boldsymbol{x}_T | \boldsymbol{x}_0, \boldsymbol{c}) = \prod_{t=1}^{T} q(\boldsymbol{x}_t | \boldsymbol{x}_{t-1}, \boldsymbol{x}_0, \boldsymbol{c})$ and the conditional reverse process $p_\theta(\boldsymbol{x}_0, \boldsymbol{x}_1, \boldsymbol{x}_2, ..., \boldsymbol{x}_T | \boldsymbol{c}) = p_\theta(\boldsymbol{x}_T | \boldsymbol{c}) \prod_{t=1}^{T} p_\theta(\boldsymbol{x}_{t-1} | \boldsymbol{x}_t, \boldsymbol{c})$, the objective function Equation (6) is an upper bound of the negative log likelihood.*

**Proof 4**

$$
\begin{aligned}
-\log p_\theta(\boldsymbol{x}_0|\boldsymbol{c}) \leq & -\log p_\theta(\boldsymbol{x}_0|\boldsymbol{c}) + \mathbb{E}_{q(\boldsymbol{x}_{1:T}|\boldsymbol{x}_0,\boldsymbol{c})}\left\{-\log\frac{p_\theta(\boldsymbol{x}_{1:T}|\boldsymbol{x}_0,\boldsymbol{c})}{q(\boldsymbol{x}_{1:T}|\boldsymbol{x}_0,\boldsymbol{c})}\right\} \\
= & \mathbb{E}_{q(\boldsymbol{x}_{1:T}|\boldsymbol{x}_0,\boldsymbol{c})}\left\{-\log\frac{p_\theta(\boldsymbol{x}_{0:T}|\boldsymbol{c})}{q(\boldsymbol{x}_{1:T}|\boldsymbol{x}_0,\boldsymbol{c})}\right\} \\
= & -\mathbb{E}_{q(\boldsymbol{x}_{1:T}|\boldsymbol{x}_0,\boldsymbol{c})}\left\{\log\frac{p_\theta(\boldsymbol{x}_T|\boldsymbol{c})\prod_{t=1}^{T}p_\theta(\boldsymbol{x}_{t-1}|\boldsymbol{x}_t,\boldsymbol{c})}{\prod_{t=1}^{T}q(\boldsymbol{x}_t|\boldsymbol{x}_{t-1},\boldsymbol{x}_0,\boldsymbol{c})}\right\} \\
= & -\mathbb{E}_{q(\boldsymbol{x}_{1:T}|\boldsymbol{x}_0,\boldsymbol{c})}\left\{\log p_\theta(\boldsymbol{x}_T|\boldsymbol{c}) + \sum_{t>1}\log\frac{p_\theta(\boldsymbol{x}_{t-1}|\boldsymbol{x}_t,\boldsymbol{c})}{q(\boldsymbol{x}_t|\boldsymbol{x}_{t-1},\boldsymbol{x}_0,\boldsymbol{c})} + \log\frac{p_\theta(\boldsymbol{x}_0|\boldsymbol{x}_1,\boldsymbol{c})}{q(\boldsymbol{x}_1|\boldsymbol{x}_0,\boldsymbol{c})}\right\} \\
= & -\mathbb{E}_{q(\boldsymbol{x}_{1:T}|\boldsymbol{x}_0,\boldsymbol{c})}\left\{\log p_\theta(\boldsymbol{x}_T|\boldsymbol{c}) + \log\frac{p_\theta(\boldsymbol{x}_0|\boldsymbol{x}_1,\boldsymbol{c})}{q(\boldsymbol{x}_1|\boldsymbol{x}_0,\boldsymbol{c})} \right. \\
& \left. + \sum_{t>1}\log\frac{p_\theta(\boldsymbol{x}_{t-1}|\boldsymbol{x}_t,\boldsymbol{c})}{q(\boldsymbol{x}_{t-1}|\boldsymbol{x}_t,\boldsymbol{x}_0,\boldsymbol{c})} * \frac{q(\boldsymbol{x}_{t-1}|\boldsymbol{x}_0,\boldsymbol{c})}{q(\boldsymbol{x}_t|\boldsymbol{x}_0,\boldsymbol{c})}\right\} \\
= & -\mathbb{E}_{q(\boldsymbol{x}_{1:T}|\boldsymbol{x}_0,\boldsymbol{c})}\left\{\log\frac{p_\theta(\boldsymbol{x}_T|\boldsymbol{c})}{q(\boldsymbol{x}_T|\boldsymbol{x}_0,\boldsymbol{c})} + \log p_\theta(\boldsymbol{x}_0|\boldsymbol{x}_1,\boldsymbol{c}) + \log\sum_{t>1}\frac{p_\theta(\boldsymbol{x}_{t-1}|\boldsymbol{x}_t,\boldsymbol{c})}{q(\boldsymbol{x}_{t-1}|\boldsymbol{x}_t,\boldsymbol{x}_0,\boldsymbol{c})}\right\} \\
= & D_{\mathrm{KL}}(q_\phi(\boldsymbol{x}_T|\boldsymbol{x}_0,\boldsymbol{c})\|p_\theta(\boldsymbol{x}_T|\boldsymbol{c})) - \mathbb{E}_{q(\boldsymbol{x}_1|\boldsymbol{x}_0,\boldsymbol{c})}\log p_\theta(\boldsymbol{x}_0|\boldsymbol{x}_1,\boldsymbol{c}) \\
& + \sum_{t>1}\mathbb{E}_{q(\boldsymbol{x}_t|\boldsymbol{x}_0,\boldsymbol{c})}D_{\mathrm{KL}}(q_\phi(\boldsymbol{x}_{t-1}|\boldsymbol{x}_t,\boldsymbol{x}_0,\boldsymbol{c})\|p_\theta(\boldsymbol{x}_{t-1}|\boldsymbol{x}_t,\boldsymbol{c}))
\end{aligned}
\tag{24}
$$

**lemma 4** *Assuming the relational network $r_\phi(x_0, c, t)$ is Lipschitz continuous, i.e., $\forall t, \exists a$ positive real number $C_t$ s.t. $\|r_\phi(x_0, c, t) - \|r_\phi(x_0', c, t)\| \leq C_t \|x_0 - x_0'\|$, then $\|f_\theta(x_t, c, t) - x_0\|_2^2$ is an upper bound of $D_{\mathrm{KL}}(q_\phi(x_{t-1}|x_t, x_0, c)\|p_\theta(x_{t-1}|x_t, c))$ after scaling.*

**Proof 5** *From the main text, we know that*

$$D_{\mathrm{KL}}(q_\phi(x_{t-1}|x_t, x_0, c)\|p_\theta(x_{t-1}|x_t, c)) \propto$$
$$\Big\|\mu_\theta(x_t, c, t) - \frac{\sqrt{\bar{\alpha}_{t-1}}\beta_t}{1 - \bar{\alpha}_t}x_0 - \frac{\sqrt{\alpha_t}(1 - \bar{\alpha}_{t-1})}{1 - \bar{\alpha}_t}(x_t - b_t(x_0, c)) - b_{t-1}(x_0, c)\Big\|_2^2, \quad (25)$$

*where $\mu_\theta(x_t, c, t)$ is the mean of $q_\theta(x_{t-1}|x_t, c)$. Here we discard a constant with respect to $x_0, x_t, c$. With our parameterization,*

$$\mu_\theta(x_t, c, t) = \frac{\sqrt{\bar{\alpha}_{t-1}}\beta_t}{1 - \bar{\alpha}_t}\hat{x}_0 - \frac{\sqrt{\alpha_t}(1 - \bar{\alpha}_{t-1})}{1 - \bar{\alpha}_t}(x_t - b_t(\hat{x}_0, c)) - b_{t-1}(\hat{x}_0, c), \quad (26)$$

*where $\hat{x}_0 = f_\theta(x_t, c, t)$. Thus the objective function can be simplified as:*

$$\Big\|\frac{\sqrt{\bar{\alpha}_{t-1}}\beta_t}{1 - \bar{\alpha}_t}(\hat{x}_0 - x_0) + \frac{\sqrt{\alpha_t}(1 - \bar{\alpha}_{t-1})}{1 - \bar{\alpha}_t}(b_t(\hat{x}_0, c) - b_t(x_0, c)) - (b_{t-1}(\hat{x}_0, c) - b_{t-1}(x_0, c))\Big\|_2$$
$$\leq \frac{\sqrt{\bar{\alpha}_{t-1}}\beta_t}{1 - \bar{\alpha}_t}\|\hat{x}_0 - x_0\|_2 + \frac{\sqrt{\alpha_t}(1 - \bar{\alpha}_{t-1})}{1 - \bar{\alpha}_t}\|b_t(\hat{x}_0, c) - b_t(x_0, c)\|_2 + \|b_{t-1}(\hat{x}_0, c) - b_{t-1}(x_0, c)\|_2$$
$$\leq \frac{\sqrt{\bar{\alpha}_{t-1}}\beta_t}{1 - \bar{\alpha}_t}\|\hat{x}_0 - x_0\|_2 + \frac{\sqrt{\alpha_t}(1 - \bar{\alpha}_{t-1})}{1 - \bar{\alpha}_t}k_t C_t\|\hat{x}_0 - x_0\|_2 + k_{t-1}C_{t-1}\|\hat{x}_0 - x_0\|_2$$
$$= \lambda_t\|f_\theta(x_t, c, t) - x_0\|_2$$
$$(27)$$

*Similar results can be proved for DDIMs by replacing the mean of posterior in DDPMs with DDIMs, defined by Equation (20), in Equation (25).*

Assume that the total diffusion step T is big enough and only a neglegible amount of noise is added to the data at the first diffusion step, then the term $D_{\mathrm{KL}}(q_\phi(x_T|x_0, c)\|p_\theta(x_T|c)) - \mathbb{E}_{q(x_1|x_0, c)} \log p_\theta(x_0|x_1, c)$ is approximately zero. Now combining Lemmas 3 and 4, we have the following proposition:

**Proposition 2** *The objective function defined in Equation (8) is an upper bound of the negative log likelihood.*

## C.3 ACHIEVING BETTER LIKELIHOOD WITH CONTEXTDIFF

Next, we show that CONTEXTDIFF is theoretically capable of achieving better likelihood compared to original DDPMs. As the exact likelihood is intractable, we aim to compare the optimal variational bounds for negative log likelihoods. The objective function of CONTEXTDIFF at time step t is $E_{q_\phi}D_{KL}(q_\phi(x_{t-1}|x_t, x_0, c)\|p_\theta(x_{t-1}|x_t, c))$, and its optimal solution is

$$\min_{\phi,\theta} \mathbb{E}_{q_\phi}D_{KL}(q_\phi(x_{t-1}|x_t, x_0, c)\|p_\theta(x_{t-1}|x_t, c))$$
$$= min_\phi[min_\theta \mathbb{E}_{q_\phi}D_{KL}(q_\phi(x_{t-1}|x_t, x_0, c)\|p_\theta(x_{t-1}|x_t, c))] \quad (28)$$
$$\leq min_\theta \mathbb{E}_{q_{\phi=0}}D_{KL}(q_{\phi=0}(x_{t-1}|x_t, x_0, c)\|p_\theta(x_{t-1}|x_t, c)),$$

where $\phi = 0$ denotes setting the adapter network identical to 0, and thus $min_\theta \mathbb{E}_{q_{\phi=0}}D_{KL}(q_{\phi=0}(x_{t-1}|x_t, x_0, c)\|p_\theta(x_{t-1}|x_t, c))$ is the optimal loss of origianl DDPMs objective at time t. Similar inequality can be obtained for t=1:

$$\min_{\phi,\theta} \mathbb{E}_{q_\phi} - \log p_\theta(x_0|x_1, c)$$
$$\leq \min_\theta \mathbb{E}_{q_{\phi=0}} - \log p_\theta(x_0|x_1, c). \quad (29)$$

As a result, we have the following inequality by summing up the objectives at all time step:

$$- \mathbb{E}_{q(\boldsymbol{x}_0)} \log p_\theta(\boldsymbol{x}_0)$$

$$\leq min_{\phi,\theta} \sum_{t>1} \mathbb{E}_{q_\phi} D_{KL}(q_\phi(\boldsymbol{x}_{t-1}|\boldsymbol{x}_t,\boldsymbol{x}_0,\boldsymbol{c})||p_\theta(\boldsymbol{x}_{t-1}|\boldsymbol{x}_t,\boldsymbol{c})) + \mathbb{E}_{q_\phi} - \log p_\theta(\boldsymbol{x}_0|\boldsymbol{x}_1,\boldsymbol{c}) + C$$

$$\leq min_\theta \sum_{t>1} \mathbb{E}_{q_{\phi=0}} D_{KL}(q_{\phi=0}(\boldsymbol{x}_{t-1}|\boldsymbol{x}_t,\boldsymbol{x}_0,\boldsymbol{c})||p_\theta(\boldsymbol{x}_{t-1}|\boldsymbol{x}_t,\boldsymbol{c})) + \mathbb{E}_{q_{\phi=0}} - \log p_\theta(\boldsymbol{x}_0|\boldsymbol{x}_1,\boldsymbol{c}) + C$$

(30)

, where $C = \mathbb{E} D_{\mathrm{KL}}(q_\phi(\boldsymbol{x}_T|\boldsymbol{x}_0,\boldsymbol{c})||p_\theta(\boldsymbol{x}_T|\boldsymbol{c}))$ is a constant defined by $\sqrt{\bar{\alpha}_T}$. Hence, CONTEXTD-IFF has a tighter bound for the NLL, and thus theoretically capable of achieving better likelihood, compared with the original DDPMs.

### C.4 BETTER EXPRESSION OF CROSS-MODAL SEMANTICS

We provide an in-depth analysis on why CONTEXTDIFF can better express the cross-modal semantics. Our analysis focuses on the case of optimal estimation, as the theoretical analysis of convergence requires understanding the non-convex optimization of neural network, which is beyond the scope of this paper. Based on objective function in Equation (8), the optimal solution of CONTEXTDIFF at time t can be expressed as

$$\arg \min_{\phi,\theta} \mathbb{E}_{q_\phi(\boldsymbol{x}_t,\boldsymbol{x}_0|\boldsymbol{c})} ||\boldsymbol{x}_0 - f_\theta(\boldsymbol{x}_t,\boldsymbol{c})||^2$$

$$= \arg \min_\phi \arg \min_\theta \mathbb{E}_{q_\phi(\boldsymbol{x}_t,\boldsymbol{x}_0|\boldsymbol{c})} ||\boldsymbol{x}_0 - f_\theta(\boldsymbol{x}_t,\boldsymbol{c})||^2$$

$$= \arg \min_\phi \mathbb{E}_{q_\phi(\boldsymbol{x}_t|\boldsymbol{c})} E_{q_\phi(\boldsymbol{x}_0|\boldsymbol{x}_t,\boldsymbol{c})} ||\boldsymbol{x}_0 - \mathbb{E}[\boldsymbol{x}_0|\boldsymbol{x}_t,\boldsymbol{c}]||_2^2$$

$$= \phi^*, \theta^*$$

(31)

since the best estimator under L2 loss is the conditional expectation. As a result, the optimal estimator of CONTEXTDIFF for $\boldsymbol{x}_0$ is

$$\mathbb{E}[\boldsymbol{x}_0|k_t \boldsymbol{r}_{\phi^*}(\boldsymbol{x}_0, c) + \sqrt{\bar{\alpha}_t}\boldsymbol{x}_0 + \sqrt{1-\bar{\alpha}_t}\epsilon, \boldsymbol{c}],$$

(32)

while existing methods that did not incorporate cross-modal contextual information in the forward process have the following optimal estimator:

$$\mathbb{E}[\boldsymbol{x}_0|\sqrt{\bar{\alpha}_t}\boldsymbol{x}_0 + \sqrt{1-\bar{\alpha}_t}\epsilon, \boldsymbol{c}].$$

(33)

Compared with existing methods, CONTEXTDIFF can explicitly utilize the cross-modal context $\boldsymbol{r}_{\phi^*}(x_0, c)$ to optimally recover the ground truth sample, and thus achieve better multimodal semantic coherence.

Furthermore, we analyze a toy example to show that CONTEXTDIFF can indeed utilize the cross-modal relations to better recover the ground truth sample. We consider the image embedding $x_0$ and text embedding $c$ that were generated with the following mechanism:

$$x_0 = \mu(c) + \sigma(c)\epsilon,$$

(34)

where $\epsilon$ is an independent standard gaussain, $\mu(c)$ and $\sigma^2(c)$ are the mean and variance of $x_0$ conditioned on $c$. We believe this simple model can capture the multimodal relationships in the embedding space, where the relevant images and text embeddings are closely aligned with each other. Then $x_t = \sqrt{\bar{\alpha}_t}x_0 + \sqrt{1-\bar{\alpha}_t}\epsilon^{'}$ is the noisy image embedding in original diffusion model. We aim to calculate and compare the optimal estimation error at time step t in original diffusion model and in CONTEXTDIFF:

$$\min_\theta \mathbb{E}||x_0 - f_\theta(x_0,c)||_2^2$$

$$= \mathbb{E}||x_0 - \mathbb{E}[x_0|x_t,c]||_2^2$$

(35)

The conditional expectation as the optimal estimator of DDPMs can be calculated as:

$$\mathbb{E}[x_0|x_t,c] = \mu(c) - Cov(x_0,x_t|c) * Var(x_t|c)^{-1}(\sqrt{\bar{\alpha}_t}\mu(c) - x_t)$$

$$= \mu(c) - \frac{\sqrt{\bar{\alpha}_t}\sigma(c)^2}{\bar{\alpha}_t\sigma(c)^2 + 1 - \bar{\alpha}_t}(\sqrt{\bar{\alpha}_t}\mu(c) - x_t)$$

As a result, we can calculate the estimation error of DDPMs:

$$\mathbb{E}||x_0 - E[x_0|x_t, c]||_2^2$$

$$= E||\sigma(c)\epsilon - \frac{\sqrt{\bar{\alpha}_t}\sigma(c)^2}{\bar{\alpha}_t\sigma(c)^2 + 1 - \bar{\alpha}_t}(\sqrt{\bar{\alpha}_t}\sigma(c)\epsilon + \sqrt{1 - \bar{\alpha}_t}\epsilon^{'})||_2^2$$

$$= d * \frac{(1 - \bar{\alpha}_t)\bar{\alpha}_t\sigma(c)^4 + \sigma^2(c)(1 - \bar{\alpha}_t)^2}{(\bar{\alpha}_t\sigma^2(c) + 1 - \bar{\alpha}_t)^2} \quad (36)$$

$$= d * \sigma(c)^2 \frac{1 - \bar{\alpha}_t}{\bar{\alpha}_t\sigma^2(c) + 1 - \bar{\alpha}_t}$$

Now we use CONTEXTDIFF with a parameterized adapter : $x_t = \sqrt{\bar{\alpha}_t}x_0 + \sqrt{1 - \bar{\alpha}_t}\epsilon^{'} + r(\phi, c, t)x_0$ , where $r(\phi, c, t)x_0$ is the adapter. We can similarly calculate the conditional mean as the optimal estimator of CONTEXTDIFF:

$$\mathbb{E}_\phi[x_0|x_t, c] = \mu(c) - \frac{\sigma^2(c)(r(\phi, c, t) + \sqrt{\bar{\alpha}_t})}{1 - \bar{\alpha}_t + (r(\phi, c, t) + \sqrt{\bar{\alpha}_t})^2\sigma^2} * ((r(\phi, c, t) + \sqrt{\bar{\alpha}_t})\mu(c) - x_t)$$

And the estimation error for a given $\phi$ in CONTEXTDIFF is:

$$\mathbb{E}||x_0 - \mathbb{E}_\phi[x_0|x_t, c]||_2^2$$

$$= \mathbb{E}||\sigma(c)\epsilon - \frac{\sigma^2(c)(r(\phi, c, t) + \sqrt{\bar{\alpha}_t})}{1 - \bar{\alpha}_t + (r(\phi, c, t) + \sqrt{\bar{\alpha}_t})^2\sigma^2(c)}((r(\phi, c, t) + \sqrt{\bar{\alpha}_t})\sigma(c)\epsilon + \sqrt{1 - \bar{\alpha}_t}\epsilon^{'})||_2^2$$

$$= d\sigma(c)^2 \frac{1 - \bar{\alpha}_t}{1 - \bar{\alpha}_t + (r(\phi, c, t) + \sqrt{\bar{\alpha}_t})^2\sigma^2(c)}$$

$$(37)$$

Comparing the denominators of two estimation errors (Equations (36) and (37)), we can see that using a non-negative adapter will always reduce the estimation error.

## D  MORE MODEL ANALYSIS

**Comparison on Computational Costs**   We compare our method with LDM and Imagen regarding parameters, training time, and testing time in Tab. 5. We find that our context-aware adapter (188M) only introduces few additional parameters and computational costs to the diffusion backbone (3000M), and substantially improves the generation performance, achieving a better trade-off than previous diffusion models.

Table 5: Comparison on Computational Costs.

| Method | #Params | #Training Cost | # Inference Cost | FID |
|---|---|---|---|---|
| LDM | 1.4B | 0.39 s/Img | 3.4 s/Img | 12.64 |
| SDXL | 10.3B | 0.71 s/Img | 9.7 s/Img | 8.32 |
| DALL·E 2 | 6.5B | 2.38 s/Img | 21.9 s/Img | 10.39 |
| Imagen | 3B | 0.79 s/Img | 13.2 s/Img | 7.27 |
| Our CONTEXTDIFF | 3B+188M | 0.83 s/Img | 13.4 s/Img | **6.48** |

**Contributing to Faster Convergence**   In the Figure 7 of main text, we generalize our context-aware adapter to other video diffusion model Tune-A-Video and make a faster and better model convergence. Here, we additionally generalize the adapter to image diffusion model LDM and plot the partial training curve on the subset of LAION dataset in Figure 12. Similar to video domain, our context-aware adapter can also substantially improve the model convergence for image diffusion models, demonstrating the effectiveness and generalization ability of our method.

**Quantitative Results on Likelihood**   We follow (Kim et al., 2022) to compute NLL/NELBO (Negaitve Log-Likelihood/Negative Evidence Lower Bound) for performances of density estimation with Bits Per Dimension (BPD). We train our context-aware adapter on CIFAR-10 and compute NLL with the uniform dequantization. As the results in Tab. 6, we conclude that our method is empirically capable of achieving better likelihood compared to original DDPMs.

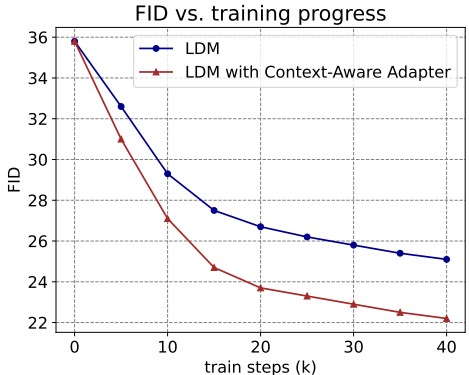

Figure 12: Our context-aware adapter can enable faster model convergence (partial training curve)
.

Table 6: NLL comparison on CIFAR-10.

| Method | NLL ↓ |
|---|---|
| DDPM (Jahn et al., 2021) | 3.01 |
| DDPM + Context-Aware Adapter | **2.63** |

## E  HYPER-PARAMETERS IN CONTEXTDIFF

We provide detailed hyper-parameters in training CONTEXTDIFF for text-to-image generation (in Tab. 7) and text-to-video editing (in Tab. 8).

Table 7: Hyper-parameters in training our CONTEXTDIFF for text-to-image generation.

| Configs/Hyper-parameters | Values |
|---|---|
| $T$ | 1000 |
| Noise schedule | cosine |
| Number of transformer blocks for cross-modal interactions | 4 |
| Betas of AdamW (Loshchilov & Hutter, 2018) | (0.9, 0.999) |
| Weight decay | 0.0 |
| Learning rate | 1$e$-4 |
| Linear warmup steps | 10000 |
| Batch size | 1024 |

## F  MORE QUALITATIVE COMPARISONS

### F.1  MORE QUALITATIVE COMPARISONS ON TEXT-TO-IMAGE GENERATION

In order to fully demonstrate the effectiveness of our proposed contextualized diffusion, we visualize more qualitative comparison results in Figure 13. The results sufficiently demonstrate the superior cross-modal understanding in generated images of our CONTEXTDIFF over other models.

### F.2  GENERALIZING TO OTHER TEXT-GUIDED VIDEO DIFFUSION MODELS

**Qualitative Results**   In order to fully demonstrate the generalization ability of the context-aware adapter in our contextualized diffusion, we visualize more qualitative comparison results, where we utilize context-aware adapter to improve Tune-A-Video (Wu et al., 2022) (in Figure 14) and FateZero (Qi et al., 2023) (in Figure 15 and Figure 16). From the results, we observe that our context-aware adapter can effectiveness promote the performance of text-to-video editing, significantly enhancing the semantic alignment while maintaining structural information in source videos. All video examples are also provided in the supplementary material, and we are committed to open sourcing the train/inference code upon paper acceptance.

Table 8: Hyper-parameters in training our CONTEXTDIFF for text-to-video editing.

| Configs/Hyper-parameters | Values |
|---|---|
| $T$ | 20 |
| Noise schedule | linear |
| $(\beta_{start}, \beta_{end})$ | $(0.00085, 0.012)$ |
| Number of transformer blocks for cross-modal interactions | 4 |
| Frames for causal attention | 3 |
| Betas of AdamW (Loshchilov & Hutter, 2018) | $(0.9, 0.999)$ |
| Weight decay | $1e$-2 |
| Learning rate | $1e$-5 |
| Warmup steps | 0 |
| Use checkpoint | True |
| Batch size | 1 |
| Number of frames | 8$\sim$24 |
| Sampling rate | 2 |

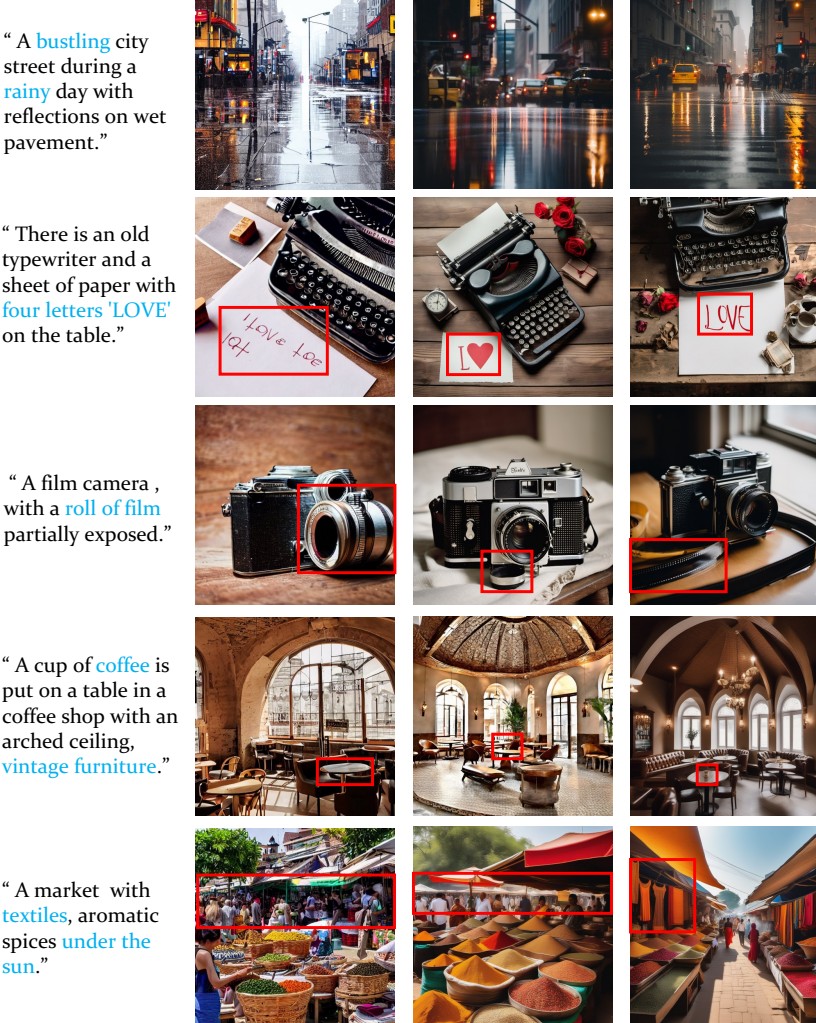

Figure 13: Synthesis examples demonstrating text-to-image capabilities of for various text prompts with LDM, Imagen, and CONTEXTDIFF (Ours). Our model can better express the semantics of the texts marked in blue. We use red boxes to highlight critical fine-grained parts where LDM and Imagen fail to align with texts. For example, in second row, only our method successfully generates the four letters spelling "LOVE". In third row, we generate the specific detail of a film roll, while other methods lose this detail.

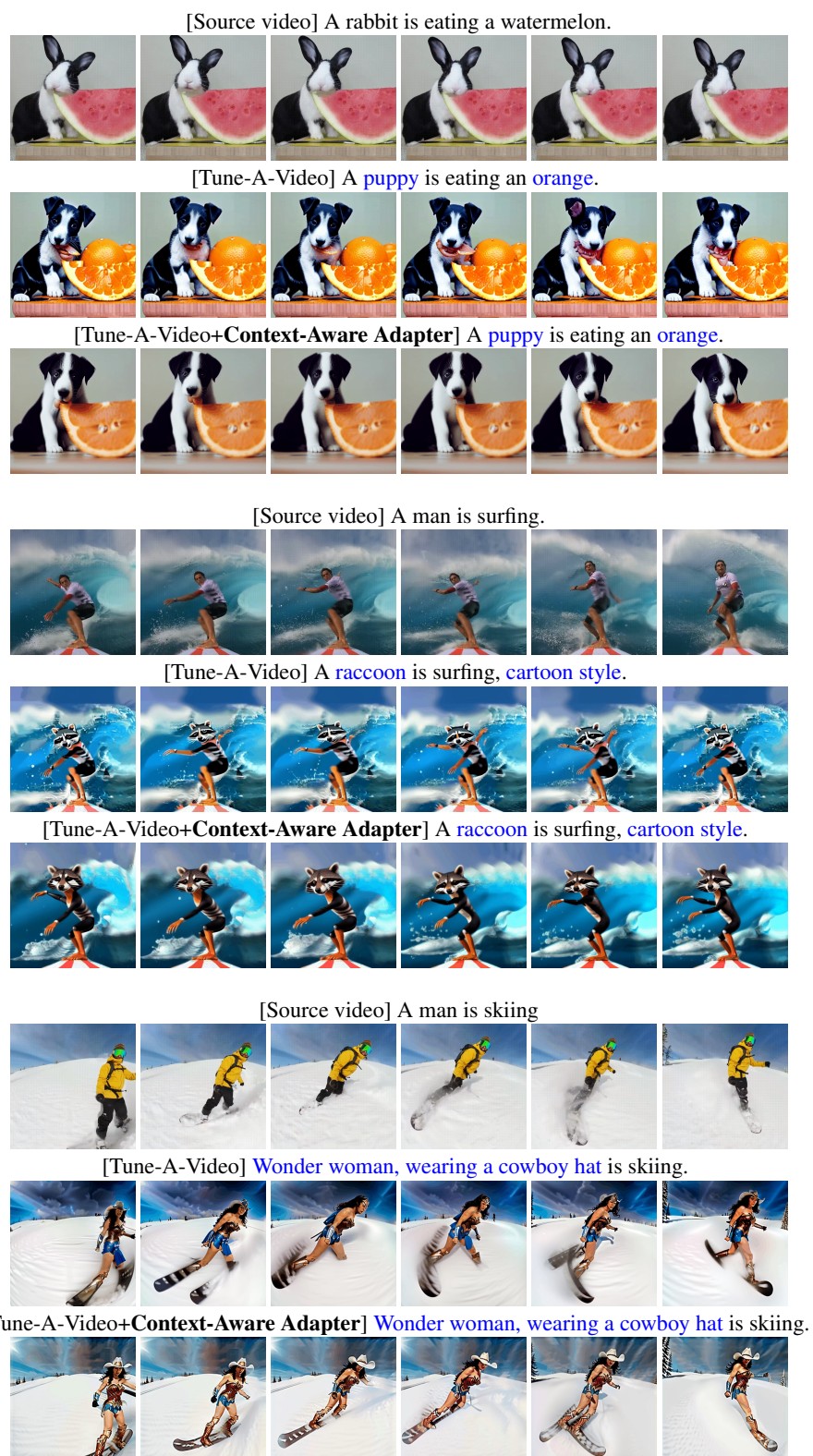

Figure 14: Generalizing our context-aware adapter to Tune-A-Video (Wu et al., 2022).

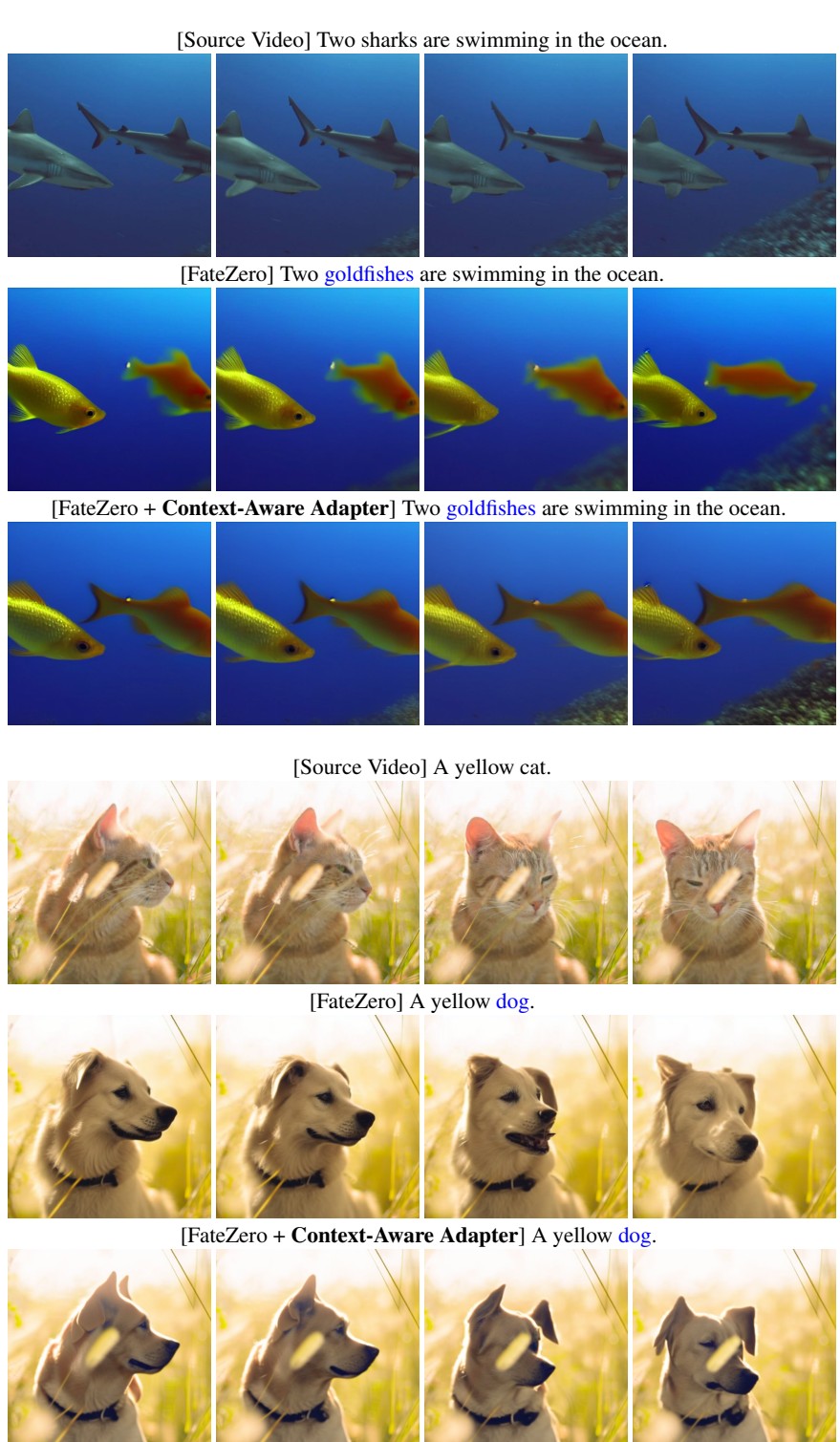

Figure 15: Generalizing our context-aware adapter to FateZero (Qi et al., 2023).

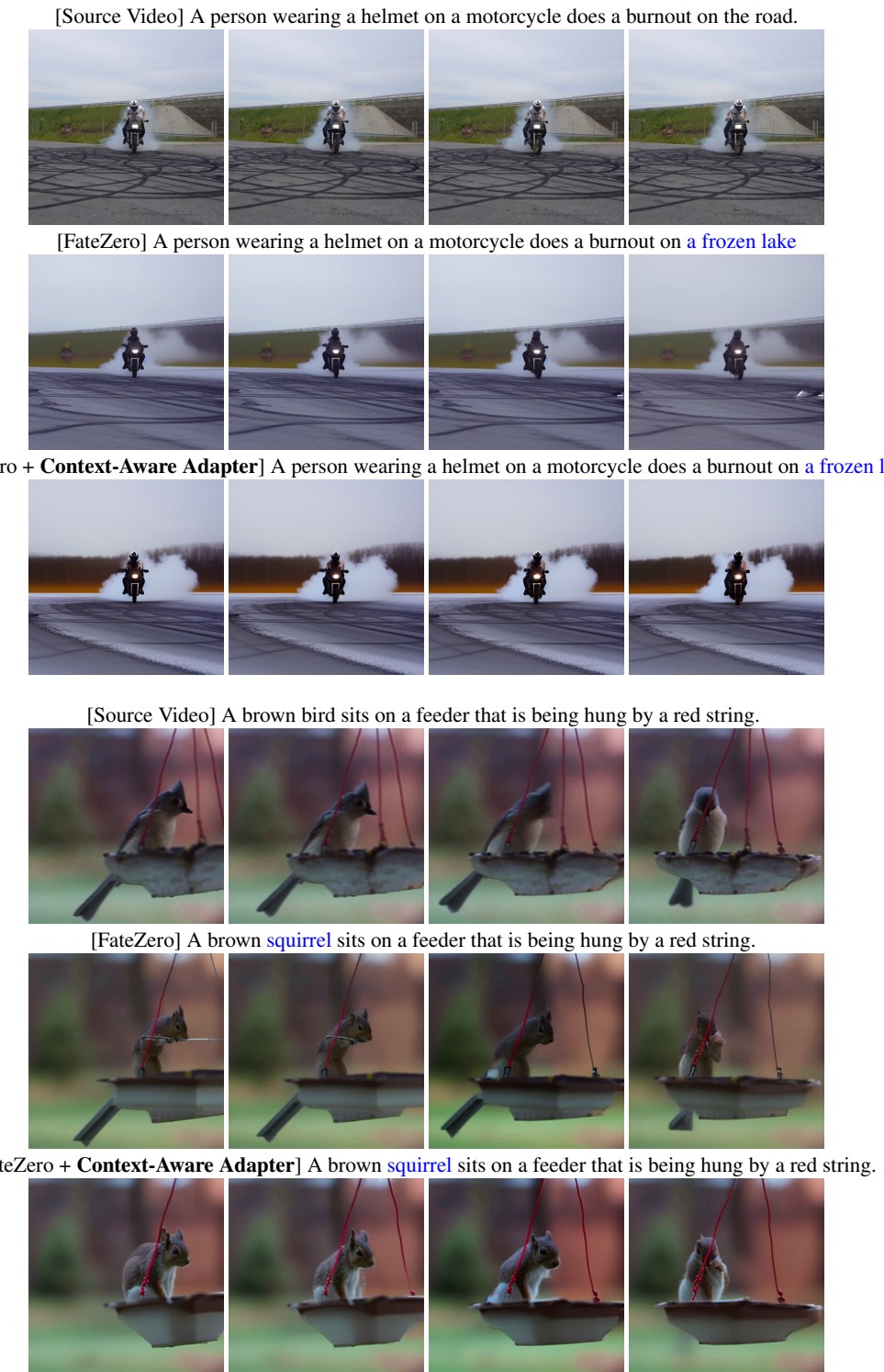

Figure 16: Generalizing our context-aware adapter to FateZero (Qi et al., 2023).

