# OpenReview forum: "Cross-Modal Contextualized Diffusion Models for Text-Guided Visual Generation and Editing"
_ICLR.cc/2024/Conference — ICLR 2024 poster_

### Official Review · Reviewer_YRdg · 2023-10-25

**Soundness:** 3 good
**Presentation:** 3 good
**Contribution:** 3 good
**Rating:** 6
**Confidence:** 4

**Summary:**

This paper proposes a contextualized diffusion model ContextDiff to facilitate the learning capacity of cross-modal diffusion models. It incorporates the cross-modal interactions between text condition and visual sample into both forward and reverse processes, serving as a context-aware adapter to optimize diffusion trajectories. It is also generalized to both DDPMs and DDIMs for benefiting both cross-modal generation and editing tasks with detailed theoretical derivations. Experiments in text-to-image generation and text-to-video editing tasks show its effectiveness. Empirical results reveal that it can successfully improve the semantic alignment between text conditions and synthesis results.

**Strengths:**

1) This paper for the first time proposes the text-guided visual diffusion model to consider cross-modal interactions in both forwarding and sampling processes.

2) The authors propose their contextualized diffusion model (ContextDiff) and generalize it to DDPMs and DDIMs through the derivations of theoretical formulas.

3) The experiments show that the proposed method has improvements on two tasks: T2I generation and T2V editing, compared with other state-of-the-art methods.

**Weaknesses:**

1) This paper claims the problem that neglecting the cross-model context in the forward process may limit the expression of textual semantics in synthesis results, but there is no clear explanation of the specific reasons, and there is no intuitive and theoretical analysis of the necessity of adding cross-model to the forward process.

2) As claimed by the authors: “Thus CONTEXTDIFF is theoretically capable of achieving better likelihood compared to original DDPMs”. Please provide the quantitative results on ELBO/ likelihood compared to the baseline.

3) Since this paper is a general improvement on the conditional diffusion model, more results on different conditional generation tasks, such as class-to-image/layout-to-image…should be provided.

4) Some configurations in T2V editing experiments are confusing, as the experiments based on pre-trained Stable Diffusion v1.4 are not enough to prove that the approach of this paper can enable diffusion models better editing ability.

**Questions:**

1) Are there any more experiments that can prove the text-based editing ability of this approach, for example, conducting T2I editing or T2V editing based on the well-trained T2I generation model of your first experiment?

---

> ### Author Response · Authors · 2023-11-16
> **Response to Reviewer YRdg (Part 1/2)**
>
> *We thank Reviewer YRdg for the positive review and valuable feedback. We are glad that the reviewer found that the proposed method is innovative, the method has theoretical foundations, and the proposed method consistently improves T2I and T2V tasks. Please see below for our responses to your comments, and **the changes in revised manuscript are marked in blue**.*
>
> **Q1: Intuitive and theoretical analysis of the necessity of adding cross-modal context to the forward process**
>
> A1:
>
> **Qualitative Analysis**: We investigate how our adapter enhances multimodal semantic relevance in the **Figure 8 and Figure 9, marked in blue**. We visualize the heatmaps of text-image cross-attention module in the sampling process of each frame image. We find that the image latents refined by our adapter better attend to the fine-grained semantics in text and sufficiently convey them for generation.
>
> **Theoretical Analysis**:
> We provide an in-depth analysis on why ContextDiff can better express the cross-modal semantics. Based on objective function in Equation (8), the optimal solution of ContextDiff at time t can be expressed as
> $$\begin{array}{rl}& \arg\min_{\phi,\theta}E_{q_\phi(x_t,x_0|c)}||x_0-f_\theta(x_t,c)||^2\\\\
> &=\arg\min_{\phi}E_{q_\phi(x_t|c)}E_{q_\phi(x_0|x_t,c)}||x_0-E[x_0|x_t,c]||_2^2 \\\\
> & = \phi^*,\theta^* \\\\
> \end{array}$$
>
>
> ,since the best estimator under L2 loss is the conditional expectation. As a result, the optimal estimator of ContextDiff for $x_0$ is
> $$E[x_0|k_t r_{\phi^*}(x_0,c)+\sqrt{\bar{\alpha}_t}x_0+\sqrt{1-\bar{\alpha}_t}\epsilon, c],$$
>
> while existing methods that did not incorporate cross-modal contextual information in the forward process have the following optimal estimator:
> $$E[x_0|\sqrt{\bar{\alpha}_t}x_0+\sqrt{1-\bar{\alpha}_t}\epsilon, c].$$
>
> Compared with existing methods, ContextDiff can explicitly utilize the cross-modal relation $r_{\phi^*}(x_0,c)$ to optimally recover the ground truth sample, and thus achieve better multimodal semantic alignment.
>
> Furthermore, we analyze a toy example to show that ContextDiff can indeed utilize the cross-modal context to better recover the ground truth sample. We consider the image embedding $x_0$ and text embedding $c$, which are generated with the following mechanism:
> $$x_0 = \mu(c)+\sigma(c)\epsilon,$$
> where $\epsilon$ is an independent standard gaussain, $\mu(c)$ and $\sigma^2(c)$ are the mean and variance of $x_0$ conditioned on $c$. We believe this simple model can capture the multimodal relationships in the embedding space, where the relevant images and text embeddings are closely aligned with each other. Then $x_t = \sqrt{\bar{\alpha}_t} x_0+\sqrt{1-\bar{\alpha}_t} \epsilon^{'}$ is the noisy image embedding in original diffusion model. We aim to calculate and compare the optimal estimation error in original diffusion model and in ContextDiff:
>
>  $$\begin{aligned}&min_\theta E_{q(x_0,x_t|c)}||x_0-f_\theta(x_0,c)||_2^2 \\\\
>  &=E ||x_0-E[x_0|x_t,c]||_2^2 \end{aligned}$$
>
> The conditional expectation as the optimal estimator of DDPMs can be calculated as
> $$\begin{aligned}E[x_0|x_t,c]
> &= \mu(c) - Cov(x_0,x_t|c)*Var(x_t|c)^{-1}(\sqrt{\bar{\alpha}_t}\mu(c)-x_t)\\\\
> &= \mu(c) - \frac{\sqrt{\bar{\alpha}_t}\sigma(c)^2}{\bar{\alpha}_t\sigma(c)^2+1-\bar{\alpha}_t}(\sqrt{\bar{\alpha}_t}\mu(c)-x_t) \end{aligned}$$
>
> As a result, we can calculate the estimation error of DDPMs:
>
> $$\begin{array}{rl} &E||x_0-E[x_0|x_t,c]||_2^2 \\\\
> &=E||\sigma(c)\epsilon-\frac{\sqrt{\bar{\alpha}_t}\sigma(c)^2}{\bar{\alpha}_t\sigma(c)^2+1-\bar{\alpha}_t}(\sqrt{\bar{\alpha}_t}\sigma(c)\epsilon+\sqrt{1-\bar{\alpha}_t} \epsilon^{'})||_2^2\\\\
> & = d*\frac{(1- \bar{\alpha}_t)\bar{\alpha}_t\sigma(c)^4+ \sigma^2(c)(1-\bar{\alpha}_t)^2}{(\bar{\alpha}_t \sigma^2(c)+1- \bar{\alpha}_t)^2}\\\\
> & = d *\sigma(c)^2 \frac{1- \bar{\alpha}_t}{\bar{\alpha}_t \sigma^2(c)+1- \bar{\alpha}_t}
>  \end{array}$$
> Now we use CONTEXTDIFF with a parameterized adapter : $x_t = \sqrt{\bar{\alpha}_t} x_0+\sqrt{1-\bar{\alpha}_t} \epsilon^{'}+r(\phi,c,t)x_0$
>
>  , where $r(\phi,c,t)x_0$ is the adapter. We can similarly calculate the conditional mean as the optimal estimator of ContextDiff:
>  $$E_\phi[x_0|x_t,c] = \mu(c)-\frac{\sigma^2(c)(r(\phi,c,t)+\sqrt{\bar{\alpha}_t})}{1-\bar{\alpha}_t+(r(\phi,c,t)+\sqrt{\bar{\alpha}_t})^2\sigma^2}*((r(\phi,c,t)+\sqrt{\bar{\alpha}_t})\mu(c)-x_t)$$
>  And the estimation error for a given $\phi$ in ContextDiff is:
>
> $$\begin{array}{rl}
> &E||x_0-E_\phi[x_0|x_t,c]||_2^2\\\\
> &=E||\sigma(c)\epsilon-\frac{\sigma^2(c)(r(\phi,c,t)+\sqrt{\bar{\alpha}_t})}{1-\bar{\alpha}_t+(r(\phi,c,t)+\sqrt{\bar{\alpha}_t})^2\sigma^2(c)}((r(\phi,c,t)+\sqrt{\bar{\alpha}_t})\sigma(c)\epsilon+\sqrt{1-\bar{\alpha}_t} \epsilon^{'})||_2^2\\\\
> &= d \sigma(c)^2\frac{1-\bar{\alpha}_t}{1-\bar{\alpha}_t+(r(\phi,c,t)+\sqrt{\bar{\alpha}_t})^2\sigma^2(c)}
>  \end{array}$$
>
> Comparing the denominators of two estimation errors, we can see that using a non-negative adapter will always reduce the estimation error.

---

> ### Author Response · Authors · 2023-11-16
> **Response to Reviewer YRdg (Part 2/2)**
>
> **Q2: Provide the quantitative results on ELBO/ likelihood compared to the baseline**
>
> A2: We compute NLL (Negaitve Log-Likelihood) for performances of density estimation with Bits Per Dimension (BPD). We train our ContextDiff on CIFAR-10 from scratch and use the same training steps as DDPM, then compute NLL with the uniform dequantization. The results are in the table below, we conclude that our method is empirically capable of achieving better likelihood compared to original DDPMs.
>
> | Method |NLL $\downarrow$|
> | :-----| :----: |
> |DDPM|3.01|
> |DDPM + Context-Aware Adapter|**2.63**|
>
> **Q3: More results on different conditional generation tasks, such as class-to-image/layout-to-image generation**
>
> A3: We generalize our context-aware adapter into class and layout conditional generation tasks. We replace the text encoder in original adapter with ResNet blocks for embedding classes or layouts, and keep the original image encoder and cross-attention module for obtaining cross-modal context information. We put both quantitative (**Table 3 and Table 4**) and qualitative results (**Figure 10 and Figure 11**) **in the updated manuscript marked in blue**. From the results, we conclude that our context-aware adapter can benefit the conditional diffusion models with different condition modalities and enable more realistic and precise generation consistent with input conditions, demonstrating the satisfying generalization ability of our method.
>
> **Q4: Are there any more experiments that can prove the text-based editing ability of this approach?**
>
> A4: In Section 5.2, we have demonstrate that our context-aware adapter can improve video diffusion models in **text-guided video editing tasks**. Kindly note that existing diffusion-based T2V editing methods (Tune-A-Video [1] and FateZero [2]) are mainly based on a pretrained T2I model (Stable Diffusion v1.4/1.5 or higher) for preliminary appearance generation ability, and they try to develop spatio-temporal approaches to improve the semantic precision of edited frames and the temporal consistency between frames. We base on pretrained Stable Diffusion v1.4 just **for fair comparison**, and both quantitative and qualitative results have already proved the text-based editing ability of our approach. For more insights, the **heatmap visualization results in Figure 8 and Figure 9** of the updated manuscript further show and explain our superior editing ability.
>
> [1] Wu J Z, Ge Y, Wang X, et al. Tune-a-video: One-shot tuning of image diffusion models for text-to-video generation. In IEEE International Conference on Computer Vision. 2023.
>
> [2] Chenyang Qi, Xiaodong Cun, Yong Zhang, Chenyang Lei, Xintao Wang, Ying Shan, and Qifeng Chen. Fatezero: Fusing attentions for zero-shot text-based video editing. In IEEE International Conference on Computer Vision, 2023.

---

> ### Comment · Reviewer_YRdg · 2023-11-20
> **Comments by Reviewer YRdg**
>
> Thanks for the feedback. The authors have met most of my concerns. I still lean toward keeping my rating as marginally above the acceptance threshold.

---

### Official Review · Reviewer_zGd6 · 2023-10-29

**Soundness:** 4 excellent
**Presentation:** 3 good
**Contribution:** 4 excellent
**Rating:** 8
**Confidence:** 3

**Summary:**

This work proposes a novel conditional diffusion model, ContextDiff. Rather than only modeling the cross-modal context in the backward process, ContextDiff propagates the context information to all timesteps in both forward and backward process to adapt the trajectories for facilitating cross-modal conditional generation. The proposed method can be generalized to DDPMs and DDIMs and achieves better results in text-to-image generation and text-to-video editing.

**Strengths:**

* The idea of modeling the cross-modal context in the forward process is interesting, as it differs from previous works that only consider conditional modeling in the backward process.
* The method has sound theoretical foundations. Furthermore, the generalization to DDIMs is a clear strength that allows fast sampling.
* The writing of the method section is clear. The adaptation to the previous diffusion process with a bias term is straightforward to understand.
* The evaluation results show that the model performs better in terms of qualitative metrics of both automated evaluation and user study.

**Weaknesses:**

* Experiments on latent diffusion: the method uses an Imagen-based framework, which generates a low-res image and then performs super-resolution. However, the author does not evaluate the proposed method on latent diffusion architecture. There are mentions of LDM in Sec 5.3 (ablations), but the setting is not clearly described, and comparisons with other works (rather than the baseline) are not offered.
* The author does not offer an inference latency evaluation. Does the method slow inference down compared to baseline diffusion methods that do not have context-aware adapters?
* A small typo (which does not affect the rating): "A red ross" -> "A red rose"?

**Questions:**

* How does the method compare to the baseline when integrated into latent diffusion (or Stable Diffusion)?
* How does the method compare to the baseline in terms of inference latency?
* What is the Stable Diffusion version used in Table 1? How does the method compare with different versions of Stable Diffusion?

---

> ### Author Response · Authors · 2023-11-16
> **Response to Reviewer zGd6**
>
> *We thank Reviewer zGd6 for the positive review and valuable feedback. We are glad that the reviewer found that the idea is interesting, the method has sound theoretical foundations, the writing is clear, and the evaluations are sufficient for demonstrating the effectiveness. Please see below for our responses to your comments, and **the changes in revised manuscript are marked in blue**.*
>
> **Q1: How does the method compare to the baseline when integrated into latent diffusion (or Stable Diffusion)**
>
> A1: Kindly note that our context-aware adapter can improve both text-guided image diffusion models and video diffusion models with latent diffusion architectures. In the ablation studies of Figure 6, we evaluate the FID-5k performance of LDM and LDM + our context-aware adapter, and the adapter is optimized with a pre-trained LDM. For fair comparison, we ensure that two models have the same number of training steps.
> From the results, we conclude that our method can comprehensively improve other pretrained latent diffusion models (i.e., Stable Diffusion). In the ablation studies of **Figure 5, 14, 15, and 16** and the quantitative results of **Table 2**, we sufficiently demonstrate our context-aware adapter can also improve other video diffusion models (Tune-A-Video and FateZero), which are all **based on latent diffusion architecture**, and outperform all previous methods. Further, we generalize our latent-based context-aware adapter to class and layout conditional image generation tasks for demonstrating the effectiveness. The results are **in the Table 3 and Table 4 of the revised manuscript marked in blue**, we find that our method not only improves the LDM model, but also surpasses other works in different conditional generation tasks.
>
> **Q2: Inference latency of proposed method and the Stable Diffusion version used for comparison**
>
> A2: We compare our method with other diffusion models regarding model size, training time, inference time and FID performance in the table below:
>
> | Method |Parameters $\downarrow$| Training Time $\downarrow$| Inference Time $\downarrow$|FID $\downarrow$|
> | :-----| :----: | :----: |:----: |:----:|
> |LDM (Stable Diffusion v1.0)|1.4B|0.39 s/Img|3.4 s/Img|12.64|
> |Stable Diffusion v1.5|0.9B|0.63 s/Img|6.5 s/Img|9.62|
> |Stable Diffusion XL|10.3B|0.71 s/Img|9.7 s/Img|8.32|
> |DALL·E 2|6.5B|2.38 s/Img|21.9 s/Img|10.39|
> |Imagen |3B| 0.79 s/Img|13.2 s/Img|7.27|
> |Imagen + Our Context-Aware Adapter| 3B+188M|0.83 s/Img|13.4 s/Img|**6.48**|
>
> We observe that our context-aware adapter (188M) only introduces few additional parameters and computational costs to the diffusion backbone (3000M), and substantially improves the generation performance, achieving a better trade-off than previous diffusion models. In Table 1, we use LDM (Stable Diffusion v1.0) for comparison, and our ContextDiff can consistently outperform different versions of Stable Diffusion as illustrated in the table above.
>
> **Q3: A small typo: "A red ross" -> "A red rose"**
>
> A3: Thanks for your reminder, we have corrected it in the revised manuscript.

---

> > ### Comment · Reviewer_zGd6 · 2023-11-22
> >
> > Thanks for the updates in the rebuttal. The comparison and clarifications with Stable Diffusion are indeed very helpful in understanding the method. I still vote for acceptance of this work. Keeping my score as is.

---

### Official Review · Reviewer_D2VS · 2023-10-30

**Soundness:** 3 good
**Presentation:** 2 fair
**Contribution:** 2 fair
**Rating:** 5
**Confidence:** 4

**Summary:**

This paper proposes a  general cross-modal contextualized diffusion model (CONTEXTDIFF) that harnesses cross-modal context to facilitate the learning capacity of cross-modal diffusion models. The cross-modal interactions between text condition and image/video sample are incorperated into the forward process, serving as a context-aware adapter to optimize diffusion trajectories. The context-aware adapter to adapt the sampling trajectories, which facilitates the conditional modeling in the reverse process and aligns it with the adapted forward process. A series of experimental results and mathematical proofs are presented.

**Strengths:**

1. The paper proposes a method to enhance the multimodal relevance by incorporating multimodal contextual information during the forward process of the diffusion model.

2. Adequate mathematical proofs are provided for both the forward and backward processes.

3. The experimental results to some extent demonstrate that this method yields a high semantic correlation between the generated images and text.

**Weaknesses:**

1. Compared to the proposed method, existing methods also utilize cross-modal attention during the forward process to control the generated content of images based on information from different modalities.

2. The article does not provide sufficient analysis for why adding textual information during the forward process enhances multimodal semantic relevance. It is based on intuitive reasoning rather than an in-depth analysis.

3. In the provided experimental results, the method proposed in this paper shows limited improvements compared to existing methods (e.g. Imagen) in image generation tasks.

4. The paper does not analyze the limitations of the proposed method.

**Questions:**

1. In Figure 6, the author claims that they conduct ablation study on the trade-off between CLIP and FID scores across a range of guidance weights, however, only FID scores are provided in the figure.

2.  Analysis on why adding textual information during the forward process enhances multimodal semantic relevance.

3. Whether this method can be incrementally trained on other pretrained generative models (e.g. Stable Diffusion), and if doing so would result in improved generation performance and faster convergence, is not discussed in the paper.

---

> ### Author Response · Authors · 2023-11-16
> **Response to Reviewer D2VS (Part 1/2)**
>
> *We thank Reviewer D2VS for the valuable feedback. We are glad that the reviewer think that we provide adequate mathematical proofs for our new method and this method yields a high semantic correlation between the generated images and text. Please see below for our responses to your comments, and **the changes in revised manuscript are marked in blue**.*
>
> **Q1: Existing diffusion-based methods utilize cross-modal attention to control the cross-modal generation conditioned on information from different modalities.**
>
> A1: We speculate that you may want to say "existing methods also utilize cross-modal attention during the **backward (generation)** process" not "during the forward process". Because, to the best of our knowledge, we are the first to incorporate cross-modal context ($c$, $x_0$) (the relationships between text embedding $c$ and image embedding $x_0$/$\hat x_0$) to adapt both forward and backward processes of diffusion models while existing methods only add text into backward process. Regarding the mechanisms of conditioning, we propose context-aware adapter to directly and effectively adjust both forward and backward trajectories while existing methods only conduct cross attention between text and noisy latent. More direct comparison is in the table below:
>
> | Method |Forward|Backward|Conditioning Mechanism|
> | :-----| :----: |:----:|:----:|
> |Existing Diffusion Models|-|only text $c$|cross attention|
> |Our ContextDiff |context ($c$,$x_0$)|context ($c$,$\hat x_0$)|directly adjust trajectories
>
> **Q2: Regarding the improvements compared to existing methods (e.g. Imagen) in image generation tasks.**
>
> A2: Our method focuses on improving the cross-modal semantic understanding of diffusion models, which is critical for both text-guided image generation and video editing.  For better understanding of qualitative results, we use **red boxes to highlight critical fine-grained parts of Fig.3 and Fig.13 in the updated paper** where our method substantially surpasses existing methods. Besides, in text-guided video editing tasks, we significantly advance video diffusion models to precisely manipulate videos according to the edited text as in Fig.4, Fig.5, Fig.14, Fig.15, and Fig.16. As for quantitative results, our FID improvement over previous methods is also significant because improving FID performance on MS-COCO is very challenging, and we achieve new SOTA results. We also improve the quantitative results in text-guided video editing tasks with automatic and human evaluations. In conclusion, we are **the first work** to generalize new diffusion models to **both text-guided image generation and video editing** tasks, and **consistently achieve new SOTA results.**
>
> **Q3: Analyze the limitations of the proposed method.**
>
> A3: While our ContextDiff boosts performance of both text-guided image and video diffusion models introducing an efficient cross-modal context-aware adapter, our models still have more trainable parameters than other types of generative models, e.g GANs. Furthermore, we need the longer inference times compared to single step generative approaches like GANs or VAEs due to the iterative diffusion sampling. However, this drawback is inherited from the underlying model class and is not a property of our ContextDiff. For future work, we will try to find more intrinsic context information to preserve for improving existing cross-modal diffusion models, and try to design a non-parametric context-aware adapter.
>
> **Q4: Regarding the ablation study on the trade-off between CLIP and FID scores across a range of guidance weights, only FID scores are provided in the figure**
>
> A4: In fact, the CLIP score is positively correlated with guidance weight in text-to-image generation, as demonstrated in DALL·E 2 and RAPHAEL. Therefore, following DALL·E 2, we plot the FID-guidance curve in Figure 6. From Figure 6, we can observe that equipped with our context-aware adapter, LDM can achieve a better FID-guidance (FID-CLIP) trade-off.
>
> **Q5: Whether this method can be incrementally trained on other pretrained generative models, and result in improved generation performance and faster convergence.**
>
> A5: Kindly note that our context-aware adapter can improve both text-guided image and video diffusion models. In the ablation studies of **Figure 6**, we have proved our method can comprehensively improve other pretrained image diffusion models (i.e., Stable Diffusion). In the ablation studies of **Figure 5, 14, 15, and 16**, we qualitatively demonstrate our method can also improve other pretrained video diffusion models. Besides, in **Figure 7**, we have proved that our context-aware adapter can lead to better video editing performance and faster convergence of other video diffusion models. Finally, for a comprehensive evaluation, we additionally prove our method can improve generation quality and enable faster convergence of other image diffusion models (**Figure 12 in the revised manuscript, marked in blue**).

---

> ### Author Response · Authors · 2023-11-16
> **Response to Reviewer D2VS (Part 2/2)**
>
> **Q6: Sufficient analysis for why adding textual information during the forward process enhances multimodal semantic relevance**
>
> A6:
>
> **Qualitative Analysis**: We first conduct qualitative analysis on how our context-aware adapter enhances multimodal semantic relevance, the visual results are in the **Figure 8 and Figure 9 of the revised manuscript, marked in blue**. Specifically, we visualize the heatmaps of text-image cross-attention module in the sampling process of each frame image. We find that the image latents refined by our context-aware adapter can better attend to the fine-grained semantics in text prompt and sufficiently convey them in final generation results.
>
> **Theoretical Analysis**:
> We provide an in-depth analysis on why ContextDiff can better express the cross-modal semantics. Our analysis focuses on the case of optimal estimation, as the theoretical analysis of convergence requires understanding the non-convex optimization of neural network, which is beyond the scope of this paper. Based on objective function in Equation (8), the optimal solution of ContextDiff at time t can be expressed as
> $$\begin{array}{rl}& \arg\min_{\phi,\theta}E_{q_\phi(x_t,x_0|c)}||x_0-f_\theta(x_t,c)||^2\\\\
> &=\arg\min_{\phi}E_{q_\phi(x_t|c)}E_{q_\phi(x_0|x_t,c)}||x_0-E[x_0|x_t,c]||_2^2 \\\\
> & = \phi^*,\theta^* \\\\
> \end{array}$$
>
>
> ,since the best estimator under L2 loss is the conditional expectation. As a result, the optimal estimator of ContextDiff for $x_0$ is
> $$E[x_0|k_t r_{\phi^*}(x_0,c)+\sqrt{\bar{\alpha}_t}x_0+\sqrt{1-\bar{\alpha}_t}\epsilon, c],$$
>
> while existing methods that did not incorporate cross-modal contextual information in the forward process have the following optimal estimator:
> $$E[x_0|\sqrt{\bar{\alpha}_t}x_0+\sqrt{1-\bar{\alpha}_t}\epsilon, c].$$
>
> Compared with existing methods, ContextDiff can explicitly utilize the cross-modal relation $r_{\phi^*}(x_0,c)$ to optimally recover the ground truth sample, and thus achieve better multimodal semantic alignment.
>
> Furthermore, we analyze a toy example to show that ContextDiff can indeed utilize the cross-modal context to better recover the ground truth sample. We consider the image embedding $x_0$ and text embedding $c$, which are generated with the following mechanism:
> $$x_0 = \mu(c)+\sigma(c)\epsilon,$$
> where $\epsilon$ is an independent standard gaussain, $\mu(c)$ and $\sigma^2(c)$ are the mean and variance of $x_0$ conditioned on $c$. We believe this simple model can capture the multimodal relationships in the embedding space, where the relevant images and text embeddings are closely aligned with each other. Then $x_t = \sqrt{\bar{\alpha}_t} x_0+\sqrt{1-\bar{\alpha}_t} \epsilon^{'}$ is the noisy image embedding in original diffusion model. We aim to calculate and compare the optimal estimation error in original diffusion model and in ContextDiff:
>
>  $$\begin{aligned}&min_\theta E_{q(x_0,x_t|c)}||x_0-f_\theta(x_0,c)||_2^2 \\\\
>  &=E ||x_0-E[x_0|x_t,c]||_2^2 \end{aligned}$$
>
> The conditional expectation as the optimal estimator of DDPMs can be calculated as
> $$\begin{aligned}E[x_0|x_t,c]
> &= \mu(c) - Cov(x_0,x_t|c)*Var(x_t|c)^{-1}(\sqrt{\bar{\alpha}_t}\mu(c)-x_t)\\\\
> &= \mu(c) - \frac{\sqrt{\bar{\alpha}_t}\sigma(c)^2}{\bar{\alpha}_t\sigma(c)^2+1-\bar{\alpha}_t}(\sqrt{\bar{\alpha}_t}\mu(c)-x_t) \end{aligned}$$
>
> As a result, we can calculate the estimation error of DDPMs:
>
> $$\begin{array}{rl} &E||x_0-E[x_0|x_t,c]||_2^2 \\\\
> &=E||\sigma(c)\epsilon-\frac{\sqrt{\bar{\alpha}_t}\sigma(c)^2}{\bar{\alpha}_t\sigma(c)^2+1-\bar{\alpha}_t}(\sqrt{\bar{\alpha}_t}\sigma(c)\epsilon+\sqrt{1-\bar{\alpha}_t} \epsilon^{'})||_2^2\\\\
> & = d*\frac{(1- \bar{\alpha}_t)\bar{\alpha}_t\sigma(c)^4+ \sigma^2(c)(1-\bar{\alpha}_t)^2}{(\bar{\alpha}_t \sigma^2(c)+1- \bar{\alpha}_t)^2}\\\\
> & = d *\sigma(c)^2 \frac{1- \bar{\alpha}_t}{\bar{\alpha}_t \sigma^2(c)+1- \bar{\alpha}_t}
>  \end{array}$$
> Now we use CONTEXTDIFF with a parameterized adapter : $x_t = \sqrt{\bar{\alpha}_t} x_0+\sqrt{1-\bar{\alpha}_t} \epsilon^{'}+r(\phi,c,t)x_0$
>
>  , where $r(\phi,c,t)x_0$ is the adapter. We can similarly calculate the conditional mean as the optimal estimator of ContextDiff:
>  $$E_\phi[x_0|x_t,c] = \mu(c)-\frac{\sigma^2(c)(r(\phi,c,t)+\sqrt{\bar{\alpha}_t})}{1-\bar{\alpha}_t+(r(\phi,c,t)+\sqrt{\bar{\alpha}_t})^2\sigma^2}*((r(\phi,c,t)+\sqrt{\bar{\alpha}_t})\mu(c)-x_t)$$
>  And the estimation error for a given $\phi$ in ContextDiff is:
>
> $$\begin{array}{rl}
> &E||x_0-E_\phi[x_0|x_t,c]||_2^2\\\\
> &=E||\sigma(c)\epsilon-\frac{\sigma^2(c)(r(\phi,c,t)+\sqrt{\bar{\alpha}_t})}{1-\bar{\alpha}_t+(r(\phi,c,t)+\sqrt{\bar{\alpha}_t})^2\sigma^2(c)}((r(\phi,c,t)+\sqrt{\bar{\alpha}_t})\sigma(c)\epsilon+\sqrt{1-\bar{\alpha}_t} \epsilon^{'})||_2^2\\\\
> &= d \sigma(c)^2\frac{1-\bar{\alpha}_t}{1-\bar{\alpha}_t+(r(\phi,c,t)+\sqrt{\bar{\alpha}_t})^2\sigma^2(c)}
>  \end{array}$$
>
> Comparing the denominators of two estimation errors, we can see that using a non-negative adapter will always reduce the estimation error.

---

> ### Author Response · Authors · 2023-11-21
> **Gentle Reminder**
>
> Dear Esteemed Reviewer,
>
> We sincerely appreciate the time and effort you dedicated to reviewing our paper. Your valuable feedbacks have been extremely beneficial. In response, we have prepared both qualitative and theoretical analysis to address your queries and concerns.
>
> As the discussion period is approaching its conclusion in two days, we kindly request, if possible, that you review our rebuttal at your convenience. Should there be any further points that require clarification or improvement, please know that we are fully committed to addressing them promptly. Thank you once again for your invaluable contribution to our research.
>
> Warm Regards,
>
> The Authors

---

> ### Author Response · Authors · 2023-11-23
>
> Dear Reviewer D2VS:
>
> We just want to reach out to you again and see if our response addresses your concern. Your comments really inspire us, and we are eager to continue discussing our work with you. Any further discussion before today's deadline would be highly appreciated!
>
> Warm Regards,
>
> The Authors

---

### Official Review · Reviewer_DNxF · 2023-11-03

**Soundness:** 4 excellent
**Presentation:** 4 excellent
**Contribution:** 3 good
**Rating:** 6
**Confidence:** 3

**Summary:**

This paper introduces the concept of contextualized forward and reverse diffusion processes, which is interesting. They have made modifications to traditional models and proposed a method that significantly improves semantic alignment. The experimental results section effectively demonstrates the efficacy of this approach. The authors provide detailed theoretical and empirical evidence, which lends strong support to this article.

**Strengths:**

- The novelty of this article lies in the introduction of a new contextualized forward and reverse diffusion processes. They have made improvements upon existing methods and provided theoretical support.
- The results presented in this paper have shown promising performance when compared to existing models, and the visualization section further supports the efficacy of this meth

**Weaknesses:**

- Regarding evaluation metrics, while existing metrics are commonly used, aligning text, especially fine-grained text content, with images requires more refined evaluation criteria. I would like to hear the authors' opinions on the need for improved evaluation metrics for more fine-grained, context-aware image and video generation. Additionally, why not incorporate human evaluation of the generated data in this context?

-  The authors have introduced some additional controls, and it would be beneficial to discuss the associated costs, such as extra parameters, training time, and testing time, to aid in understanding their proposed method.

- The authors have achieved promising results in natural images or videos, but there is a need for further discussion regarding more fine-grained context-awareness. For instance, it would be interesting to explore whether the method remains effective when dealing with specific text from within an image, as generating precise text remains a challenge for most methods. Similarly, what happens when modifying specific parts of an image? I would like to see the authors' insights on these issues concerning their context-aware adapter.

**Questions:**

The questions I would like the authors to address have already been raised in the "weakness" section. I hope the authors can provide more information on these aspects.

---

> ### Author Response · Authors · 2023-11-16
> **Response to Reviewer DNxF**
>
> *We thank Reviewer DNxF for the positive review and valuable feedback. We are glad that the reviewer found that the proposed method is novel, performance improvements are promising, theoretical and empirical evidences are detailed and strong. Please see below for our responses to your comments, and **the changes in revised manuscript are marked in blue**.*
>
> **Q1: Opinions on the need for improved evaluation metrics for more fine-grained, context-aware image and video generation. Why not incorporate human evaluation of the generated data in this context?**
>
> A1: This is a great question for existing text-guided visual generation models, because some fine-grained text contents are often neglected in generation process. Actually, in qualitative comparison, we have noticed this problem and demonstrated our superiority in conveying fine-grained text contents as evidenced by **Fig.3 and Fig.13**. We further update the manuscript and use **red boxes to highlight critical fine-grained parts** where existing methods fail. As for better evaluation metrics, we think a comprehensive benchmark is needed to evaluate the model capacity from different aspects. For example, we need a new text prompt set categorized with color, shape, texture, spatial relationships, non-spatial relationships, and complex compositions for more comprehensive evaluation. We will attempt to build such a benchmark in future work to promote the development of this field. Following your suggestion, we conduct a preliminary fine-grained human evaluation with four kinds of fine-grained text prompts. For each kind, we try two various prompts: color ("blue" and "red"), spatial relationships ("on the left of" and "on the right of"), non-spatial relationships ("watch" and "walk with"), and complex compositions (color+spatial relationships and color+non-spatial relationships). We show the preference rate of 100 generated images with 5 subjects in the following:
>
> | Method |Color| Spatial Relationships| Non-Spatial Relationships|Complex Compositions|
> | :-----| :----: | :----: |:----: |:----: |
> |LDM | 28% | 20% | 16% |14%|
> |Imagen | 34% | 37% | 33% |32%|
> |Our ContextDiff|**38%**|**43%**|**51%**|**54%**|
>
> We conclude that our ContextDiff consistently outperforms LDM and Imagen across various fine-grained prompts, especially in complex scenarios, demonstrating the superior cross-modal understanding of our model.
>
> **Q2: Discuss the associated costs, such as extra parameters, training time, and testing time, to aid in understanding the proposed method.**
>
> A2: We compare our method with LDM and Imagen regarding parameters, training time, and testing time in the table below:
>
> | Method |Parameters $\downarrow$| Training Time $\downarrow$| Inference Time $\downarrow$|FID $\downarrow$|
> | :-----| :----: | :----: |:----: |:----:|
> |LDM|1.4B|0.39 s/Img|3.4 s/Img|12.64|
> |SDXL|10.3B|0.71 s/Img|9.7 s/Img|8.32|
> |DALL·E 2|6.5B|2.38 s/Img|21.9 s/Img|10.39|
> |Imagen |3B| 0.79 s/Img|13.2 s/Img|7.27|
> |Our ContextDiff | 3B+188M|0.83 s/Img|13.4 s/Img|**6.48**|
>
> We can observe that our context-aware adapter (188M) only introduces few additional parameters and computational costs to the diffusion backbone (3000M), and substantially improves the generation performance, achieving a better trade-off than previous diffusion models.
>
> **Q3: I would like to see the authors' insights on these issues concerning their context-aware adapter.**
>
> A3: Our ContextDiff is good at dealing with specific text within an image or modifying specific parts of an image as illustrated in our text-guided video editing results (Fig.4, Fig.5, Fig.14, Fig.15, and Fig.16), because we need to edit each frame image according to text prompts. To further investigate more fine-grained context-awareness of our model, following your suggestion, we base on our text-to-video editing framework to explore how our context-aware adapter works. We visualize the heatmaps of text-image cross-attention in the sampling process of **each frame image in Fig.8 and Fig.9 of the updated paper**. We find that our context-aware adapter can enable the model to better focus on the fine-grained text semantics and precisely modify specific parts in frame image according to the edited text prompt. The results demonstrate that our context-aware adapter can substantially improve the cross-modal understanding of the diffusion model.

---

> > ### Author Response · Authors · 2023-11-23
> > **Gentle Reminder**
> >
> > Dear Esteemed Reviewer,
> >
> > We sincerely appreciate the time and effort you dedicated to reviewing our paper. Your valuable feedbacks have been extremely beneficial. In response, we have prepared  provide more information on fine-grained evaluation metrics, computational costs, and insights about our context-aware adapter to address your queries and concerns.
> >
> > As the discussion period is approaching its conclusion, we kindly request, if possible, that you review our rebuttal at your convenience. Should there be any further points that require clarification or improvement, please know that we are fully committed to addressing them promptly. Thank you once again for your invaluable contribution to our research.
> >
> > Warm Regards,
> >
> > The Authors

---

> > > ### Comment · Reviewer_DNxF · 2023-11-23
> > > **Thanks for the responses.**
> > >
> > > Thanks for the responses of the authors. Most of them have been addressed and I will keep my original rating of borderline accept.

---

### Author Response · Authors · 2023-11-16
**General Response**

We sincerely thank all the reviewers for the thorough reviews and valuable feedback. We are glad to hear that the idea is interesting and novel (Reviewer DNxF, zGd6, and YRdg), this paper provides sound and adequate theoretical proofs for the proposed method (All Reviewers), and performance improvements showed in experiments are promising with both qualitative and quantitative results (Reviewer DNxF, zGd6, and YRdg).

We have revised the manuscript according to the suggestions of reviewers (**mainly in the appendix part, marked in blue**), we also summarized the changes of manuscript and our responses to reviewers as follows:

* We additionally provide in-depth qualitative and theoretical analysis of the necessity of adding cross-modal context to the forward process (Reviewer YRdg, DNxF and D2VS).

* We generalize our ContextDiff to more conditional image generation tasks such as class-to-image generation and layout-to-image generation for demonstrating our generalization ability (Reviewer YRdg and zGd6).

* We compare our method with other diffusion models regarding model size, training time, inference time and FID performance, achieving a better trade-off than previous diffusion models (Reviewer DNxF and zGd6).

* Our ContextDiff improves both text-guided image and video diffusion models, we make more detailed explanations for better understanding of our improvements (Reviewer D2VS, zGd6 and YRdg).

We reply to each reviewer's questions in detail below their reviews. Please kindly check out them. Thank you and please feel free to ask any further question.

---

### Meta-Review · Area_Chair_mNjf · 2023-12-17

**Metareview:**

This paper introduces contextualized forward and reverse diffusion processes and shows better performance in both text-to-image generation and video editing field. All reviewers agree this paper provides sound and adequate theoretical proofs. During rebuttal stage, authors generalize  ContextDiff to more conditional image generation task, compare our method with other diffusion models regarding model size, training time, inference time and FID performance, and more explanations. Even reviewer D2VS gives negation comment but have not involved in the discussion, the issues have been addressed. The ACs decided to accept it.

**Justification For Why Not Higher Score:**

The contextual injection into forward process is interesting, but not a significant change in SD pipeline.

**Justification For Why Not Lower Score:**

The model is novel and effective.

---

### Decision · Program_Chairs · 2024-01-16

Accept (poster)